# Average gradient outer product as a mechanism for deep neural collapse

**Daniel Beaglehole**[*,1]    **Peter Súkeník**[*,2]    **Marco Mondelli**[2]    **Mikhail Belkin**[1]

[1]UC San Diego    [2]Institute of Science and Technology Austria

## Abstract

Deep Neural Collapse (DNC) refers to the surprisingly rigid structure of the data representations in the final layers of Deep Neural Networks (DNNs). Though the phenomenon has been measured in a variety of settings, its emergence is typically explained via data-agnostic approaches, such as the unconstrained features model. In this work, we introduce a data-dependent setting where DNC forms due to feature learning through the average gradient outer product (AGOP). The AGOP is defined with respect to a learned predictor and is equal to the uncentered covariance matrix of its input-output gradients averaged over the training dataset. The Deep Recursive Feature Machine (Deep RFM) is a method that constructs a neural network by iteratively mapping the data with the AGOP and applying an untrained random feature map. We demonstrate empirically that DNC occurs in Deep RFM across standard settings as a consequence of the projection with the AGOP matrix computed at each layer. Further, we theoretically explain DNC in Deep RFM in an asymptotic setting and as a result of kernel learning. We then provide evidence that this mechanism holds for neural networks more generally. In particular, we show that the right singular vectors and values of the weights can be responsible for the majority of within-class variability collapse for DNNs trained in the feature learning regime. As observed in recent work, this singular structure is highly correlated with that of the AGOP.

## 1 Introduction

How Deep Neural Networks (DNNs) learn a transformation of the data to form a prediction and what are the properties of this transformation constitute fundamental questions in the theory of deep learning. A promising avenue to understand the hidden mechanisms of DNNs is Neural Collapse (NC) [Papyan et al., 2020]. NC is a widely observed structural property of overparametrized DNNs, occurring in the terminal phase of gradient descent training on classification tasks. This property is linked with the performance of DNNs, such as generalization and robustness [Papyan et al., 2020, Su et al., 2023]. NC is defined by four properties that describe the geometry of feature representations of the training data in the last layer. Of these, in this paper we focus on the following two, because they are relevant in the context of deeper layers. The *within-class variability collapse* (NC1) states that feature vectors of the training samples within a single class collapse to the common class-mean. The *orthogonality or simplex equiangular tight frame property* (NC2) states that these class-means form an orthogonal or simplex equiangular tight frame. While initially observed in just the final hidden layer, these properties were measured for the intermediate layers of DNNs as well [Rangamani et al., 2023, He and Su, 2022], indicating that NC progresses depth-wise. This has lead to the study of Deep Neural Collapse (DNC), a direct depth-wise extension of standard NC [Súkeník et al., 2023].

---

[*]Equal contribution. Correspondence to: Daniel Beaglehole (`dbeaglehole@ucsd.edu`), Peter Súkeník (`peter.sukenik@ista.ac.at`).

38th Conference on Neural Information Processing Systems (NeurIPS 2024).

The theoretical explanations of the formation of (D)NC have mostly relied on a data-agnostic model, known as *unconstrained features model* (UFM) [Mixon et al., 2020, Lu and Steinerberger, 2022]. This assumes that DNNs are infinitely expressive and, thus, optimizes for the feature vectors in the last layer directly. The deep UFM (DUFM) was then introduced to account for more layers [Súkeník et al., 2023, Tirer and Bruna, 2022]. The (D)UFM has been identified as a useful analogy for neural network collapse, as (D)NC emerges naturally in this setting for a variety of loss functions and training regimes (see Section 2). However, (D)UFM discards the training data and most of the network completely, thus ignoring also the role of learning in DNC formation. This is a serious gap to fully understand the formation of DNC in the context of the full DNN pipeline.

In this paper, we introduce a setting where DNC forms through a learning algorithm that is highly dependent on the training data and predictor - iterated linear mapping onto the average gradient outer product (AGOP). The AGOP is the uncentered covariance matrix of the input-output gradients of a predictor averaged over its training data. Recent work has utilized this object to understand various surprising phenomena in neural networks including grokking, lottery tickets, simplicity bias, and adversarial examples [Radhakrishnan et al., 2024a]. Additional work incorporated layer-wise linear transformation with the AGOP into kernel machines, to model the deep feature learning mechanism of neural networks [Beaglehole et al., 2023]. The output of their backpropagation-free method, Deep Recursive Feature Machines (Deep RFM), is a standard neural network at i.i.d. initialization, where each random weight matrix is additionally right-multiplied by the AGOP of a kernel machine trained on the input to that layer. Their method was shown to improve performance of convolutional kernels on vision datasets and recreate the edge detection ability of convolutional neural networks.

Strikingly, we show that the neural network generated by this very same method, Deep RFM, consistently exhibits standard DNC with little modification (Section 4). We establish in this setting that projection onto the AGOP is responsible for DNC in Deep RFM both empirically and theoretically. We verify these claims with an extensive experimental evaluation, which demonstrates a consistent formation of DNC in Deep RFM across several vision datasets. We then provide theoretical analyses that explain the mechanism for Deep RFM in the asymptotic high-dimensional regime and derive NC formation as a consequence of kernel learning (Section 5). Our first analysis is primarily based on the approximate linearity of kernel matrices when the dimension and the number of data points are large and proportional. Our second analysis demonstrates that DNC is an implicit bias of optimizing over the choice of kernel and its regression coefficients.

We then give substantial evidence that projection onto the AGOP is closely related to DNC formation in standard DNNs. In particular, we show that within-class variability for DNNs trained using SGD with small initialization is primarily reduced by the application of the right singular vectors of the weight matrix and the subsequent multiplication with its singular values (Section 6). These singular structures of a weight matrix $W$ are fully deducible from the Gram matrix of the weights at each layer, $W^\top W$. Radhakrishnan et al. [2024a], Beaglehole et al. [2023] have identified that, in many settings and across all layers of the network, $W^\top W$ is highly correlated with the average gradient outer product (AGOP) with respect to the inputs to that layer, in a statement termed the *Neural Feature Ansatz* (NFA). Thus, the NFA suggests neural networks extract features from the data representations at every layer by projection onto the AGOP with respect to that layer.

Our results demonstrate that *(i)* AGOP is a mechanism for DNC in Deep RFM, and *(ii)* when the NFA holds, i.e. with small initialization, the right singular structure of $W$, and therefore the AGOP, induces the majority of within-class variability collapse in DNNs. We thus establish projection onto the AGOP as a setting for DNC formation that incorporates the data through feature learning.

## 2    Related work

**Neural collapse (NC).**    Since the seminal paper by Papyan et al. [2020], the phenomenon of neural collapse has attracted significant attention. Galanti et al. [2022] use NC to improve generalization bounds for transfer learning, while Wang et al. [2023] discuss transferability capabilities of pre-trained models based on their NC on the target distribution. Haas et al. [2022] argue that NC can lead to improved OOD detection. Connections between robustness and NC are discussed in Su et al. [2023]. For a survey, we refer the interested reader to Kothapalli [2023].

The main theoretical framework to study the NC is the Unconstrained Features Model (UFM) [Mixon et al., 2020, Lu and Steinerberger, 2022]. Under this framework, many works have shown the emergence of the NC, either via optimality and/or loss landscape analysis [Weinan and Wojtowytsch, 2022, Lu and Steinerberger, 2022, Zhou et al., 2022], or via the study of gradient-based optimization

[Ji et al., 2022, Han et al., 2022, Wang et al., 2022]. Some papers also focus on the generalized class-imbalanced setting, where the NC does not emerge in its original formulation [Thrampoulidis et al., 2022, Fang et al., 2021, Hong and Ling, 2023]. Deviating from the strong assumptions of the UFM model, a line of work focuses on gradient descent in homogeneous networks [Poggio and Liao, 2020, Xu et al., 2023, Kunin et al., 2022], while others introduce perturbations [Tirer et al., 2022].

**AGOP feature learning.** The NFA was shown to capture many aspects of feature learning in neural networks including improved performance and interesting structural properties. The initial work on the NFA connects it to the emergence of spurious features and simplicity bias, why pruning DNNs may increase performance, the "lottery ticket hypothesis", and a mechanism for grokking in vision settings [Radhakrishnan et al., 2024a]. Zhu et al. [2023] connect large learning rates and catapults in neural network training to better alignment of the AGOP with the true features. Radhakrishnan et al. [2024b] demonstrate that the AGOP recovers the implicit bias of deep linear networks toward low-rank solutions in matrix completion. Beaglehole et al. [2024] identify that the NFA is characterized by alignment of the weights to the pre-activation tangent kernel.

# 3 Background and definitions

## 3.1 Notation

For simplicity of description, we assume a class-balanced setting, where $N = Kn$ is the total number of training samples, with $K$ being the number of classes and $n$ the number of samples per class. Note however that our experimental results and asymptotic theory hold in the general case that the classes are of unequal size. We will in general order the training samples such that the samples of the same class are grouped into blocks. With this ordering, the labels $y \in \mathbb{R}^{K \times N}$ can be written as $I_K \otimes \mathbf{1}_n^\top$, where $I_K$ denotes a $K \times K$ identity matrix, $\otimes$ denotes the Kronecker product and $\mathbf{1}_n$ a row vector of all-ones of length $n$.

For a matrix $A \in \mathbb{R}^{d_1 \times d_2}$ and a column vector $v \in \mathbb{R}^{d_1 \times 1}$, the operation $A \oslash v$, divides all elements of each row of $A$ by the corresponding element of $v$. We define the norm of $A \in \mathbb{R}^{d \times n}$, $\|A\| \in \mathbb{R}^{n \times 1}$, as the column-wise $\ell_2$-norm and not the matrix norm.

Both a DNN and Deep RFM of depth $L$ can be written as:

$$f(x) = m_{L+1}\sigma(m_L\sigma(m_{L-1}\ldots\sigma(m_1(x))\ldots)),$$

where $m_l$ is an affine map and $\sigma$ is a non-linear activation function. For neural networks, the linear map of $m_l$ is the application of a single weight matrix $W_l$, while for Deep RFM the linear transformation is a product of a weight matrix and a feature matrix $M_l^{1/2}$, written $W_l M_l^{1/2}$. The training data $X \in \mathbb{R}^{d_1 \times N}$ is stacked into columns and we let $X_l$ be the feature representations of the training data after $l$ layers of a DNN or Deep RFM before the linear layer for $l \geqslant 1$, with $X_1 = X$.

## 3.2 Average gradient outer product

The AGOP operator acts with respect to a dataset $X \in \mathbb{R}^{d_0 \times N}$ and any model, that accepts inputs from the data domain $f : \mathbb{R}^{d_0 \times 1} \to \mathbb{R}^{K \times 1}$, where $K$ is the number of outputs. Writing the (transposed) Jacobian of the model outputs with respect to its inputs as $\frac{\partial f(x)}{\partial x} \in \mathbb{R}^{d_0 \times K}$, the AGOP is defined as:

$$\text{AGOP}(f, X) \triangleq \frac{1}{N}\sum_{c=1}^{K}\sum_{i=1}^{N}\frac{\partial f(x_{ci})}{\partial x}\frac{\partial f(x_{ci})}{\partial x}^\top. \tag{1}$$

Note while the AGOP is stated such that the derivative is with respect to the immediate model inputs $x$, we will also consider the AGOP where derivatives are taken with respect to intermediate representations of the model. This object has important implications for learning because the AGOP of a learned predictor will (with surprisingly few samples) resemble the *expected* gradient outer product (EGOP) of the target function [Yuan et al., 2023]. While the AGOP is specific to a model and training data, the EGOP is determined by the population data distribution and the function to be estimated, and contains specific useful information such as low-rank structure, that improves prediction [Trivedi et al., 2014].

Remarkably, it was identified in Radhakrishnan et al. [2024a], Beaglehole et al. [2023] that neural networks will automatically contain AGOP structure in the Gram matrix of the weights in all layers

of the neural network, where the AGOP at each layer acts on the sub-network and feature vectors at that layer. Stated formally, the authors observe and pose the Neural Feature Ansatz (NFA):

**Ansatz 3.1** (Neural Feature Ansatz [Radhakrishnan et al., 2024a]). *Let $f$ be a depth-$L$ neural network trained on data $X$ using stochastic gradient descent. Then, for all layers $l \in [L]$,*

$$\rho\left(W_l^\top W_l, \frac{1}{N}\sum_{c=1}^{K}\sum_{i=1}^{n} \frac{\partial f(x_{ci})}{\partial x^l} \frac{\partial f(x_{ci})}{\partial x^l}^\top\right) \approx 1. \tag{2}$$

The second argument is the AGOP of the neural network where the gradients are taken with respect to the activations at layer $l$ and not the initial inputs. The correlation function $\rho$ accepts two matrix inputs of shape $(p, q)$ and returns the cosine similarity of the matrices flattened into vectors of length $pq$, whose value is in $[-1, 1]$. Note Radhakrishnan et al. [2024a] formulates this similarity as proportionality, which is equivalent to correlation exactly equal to 1.

Crucially, the AGOP can be defined for any differentiable estimator, not necessarily a neural network. In Radhakrishnan et al. [2024a], the authors introduce a kernel learning algorithm, the Recursive Feature Machine (RFM), that performs kernel regression and estimates the AGOP of the kernel machine in an alternating fashion, enabling recursive feature learning and refinement through AGOP.

### 3.3 Deep RFM

Subsequent work introduced the Deep Recursive Feature Machine (Deep RFM) to model deep feature learning in neural networks [Beaglehole et al., 2023]. Deep RFM iteratively generates representations by mapping the input to that layer with the AGOP of the model w.r.t. this input, and then applying an untrained random feature map. To define the Deep RFM, let $\{k_l\}_{l=1}^{L+1} : \mathbb{R}^{d_l \times d_l} \to \mathbb{R}$ be a set of kernels. For the ease of notation, we will write $k_l(X_l', X_l)$ for a matrix of kernel evaluations between columns of $X_l'$ and $X_l$. We then describe Deep RFM in Algorithm 1.

---

**Algorithm 1** Deep Recursive Feature Machine (Deep RFM)

---

**input** $X_1, Y, \{k_l\}_{l=1}^{L+1}, L, \{\Phi_l\}_{l=1}^{L}$ {kernels: $\{k_l\}_l$, depth: $L$, feature maps: $\{\Phi_l\}_l$, ridge: $\gamma$}
**output** $\alpha_{L+1}, \{M_l\}_{l=1}^{L}$
    **for** $l \in 1, \dots, L$ **do**
        Normalize the data, $X_l \leftarrow X_l \oslash \|X_l\|$
        Learn coefficients, $\alpha_l = Y(k_l(X_l, X_l) + \gamma I)^{-1}$.
        Construct predictor, $f_l(\cdot) = \alpha_l k_l(X_l, \cdot)$.
        Compute AGOP: $M_l = \sum_{c,i=1}^{K,n} \frac{\partial f_l(x_{ci}^l)}{\partial x^l} \frac{\partial f_l(x_{ci}^l)}{\partial x^l}^\top$.
        Transform the data $X_{l+1} \leftarrow \Phi_l(M_l^{1/2} X_l)$.
    **end for**
    Normalize the data, $X_{L+1} \leftarrow X_{L+1} \oslash \|X_{L+1}\|$
    Learn coefficients,
    $\alpha_{L+1} = Y(k_{L+1}(X_{L+1}, X_{L+1}) + \gamma I)^{-1}$.

---

Note that the Deep RFM as defined here considers only one AGOP estimation step in the inner-optimization of RFM, while multiple iterations are used in Beaglehole et al. [2023]. The high-dimensional feature maps $\Phi_l(\cdot)$ are usually realized as $\sigma(W_l \cdot + b_l)$, where $W_l$ is a matrix with standard Gaussian or uniform entries, $b_l$ is an optional bias term initialized uniformly at random, and $\sigma$ is the ReLU or cosine activation function. Thus, $\Phi_l$ typically serves as a random features generator.

The single loop in the Deep RFM represents a reduced RFM learning process. The RFM itself is based on kernel ridge regression, therefore we introduce the ridge parameter $\gamma$.

### 3.4 Deep Neural Collapse

We define the feature vectors $x_{ci}^l$ of the $i$-th sample of the $c$-th class as the input to $m_l$. For neural networks, we define $\tilde{x}_{ci}^l$ as the feature vectors produced from $m_{l-1}$ before the application of $\sigma$, where $l \geqslant 2$. For Deep RFM, we define $\tilde{x}_{ci}^l$ as the feature vectors produced by the application of AGOP, i.e.

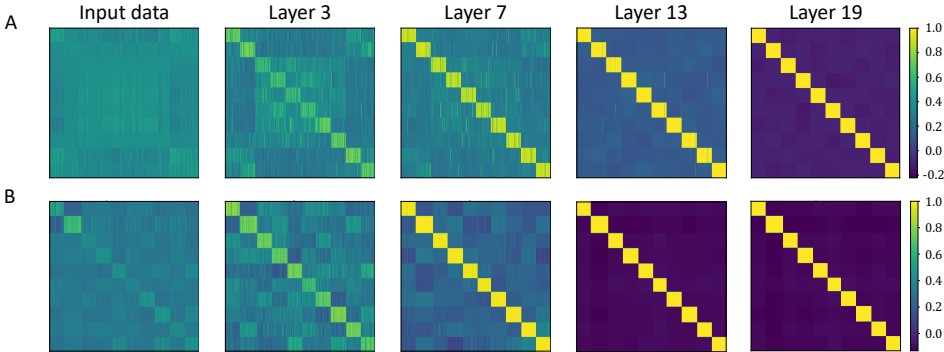

Figure 1: Neural collapse with Deep RFM on (A) CIFAR-10 and (B) MNIST. The matrix of inner products of all pairs of points in $X_l$ extracted from layers $l \in \{1, 3, 7, 13, 19\}$ of Deep RFM. The columns show the Gram matrices of feature vectors transformed by the AGOP from Deep RFM, $\left( \tilde{X}_l - \tilde{\mu}_G^l \right) \oslash \| \tilde{X}_l - \tilde{\mu}_G^l \|$. The data are ordered so that points of the same class are adjacent to one another, arranged from classes 1 to 10. Deep RFM uses non-linearity $\sigma(\cdot) = \cos(\cdot)$ in (A) and $\sigma(\cdot) = \mathrm{ReLU}(\cdot)$ in (B).

$\tilde{x}_{ci}^l = M_l^{1/2} x_{ci}^l$, again where $l \geqslant 2$. In both model types, we let $\tilde{x}_{ci}^1 = x_{ci}^1$. Let $\mu_c^l := \frac{1}{n} \sum_{i=1}^n x_{ci}^l$, $\tilde{\mu}_c^l := \frac{1}{n} \sum_{i=1}^n \tilde{x}_{ci}^l$ be the corresponding class means, and $\mu^l, \tilde{\mu}^l \in \mathbb{R}^{d_l \times K}$ the matrices of class means. NC1 is defined in terms of the *within-class* and *between-class* variability at layer $l$:

$$\Sigma_W^l = \frac{1}{N} \sum_{c=1}^K \sum_{i=1}^n (x_{ci}^l - \mu_c^l)(x_{ci}^l - \mu_c^l)^\top, \qquad \Sigma_B^l = \frac{1}{K} \sum_{c=1}^K (\mu_c^l - \mu_G^l)(\mu_c^l - \mu_G^l)^\top .$$

**Definition 3.2.** In the context of this work, NC is achieved at layer $l$ if the following two properties are satisfied:

- **DNC1:** The within-class variability at layer $l$ is 0. This property can be stated for the features either after or before the application of the activation function $\sigma$. In the former case, the condition requires $x_{ci}^l = x_{cj}^l$ for all $i, j \in \{1, \ldots, n\} \triangleq [n]$ (or, in matrix notation, $X_l = \mu^l \otimes \mathbf{1}_n^\top$); in the latter, $\tilde{x}_{ci}^l = \tilde{x}_{cj}^l$ for all $i, j \in [n]$ (or, in matrix notation, $\tilde{X}_l = \tilde{\mu}^l \otimes \mathbf{1}_n^\top$).

- **DNC2:** The class-means at the $l$-th layer form either an orthogonal basis or a simplex equiangular tight frame (ETF). As for DNC1, this property can be stated for features after or before $\sigma$. We write the ETF class-mean covariance $\Sigma_{\mathrm{ETF}} \triangleq (1 + \frac{1}{K-1})I_K - \frac{1}{K-1}\mathbf{1}_K\mathbf{1}_K^\top$. In the former case, the condition requires either $(\mu^l)^\top \mu^l \propto I_K$ or $(\mu^l)^\top \mu^l \propto \Sigma_{\mathrm{ETF}}$; in the latter $(\tilde{\mu}^l)^\top \tilde{\mu}^l \propto I_K$ or $(\tilde{\mu}^l)^\top \tilde{\mu}^l \propto \Sigma_{\mathrm{ETF}}$. In this work, we will measure this condition on the centered and normalized class means, $\bar{\mu}^l$. We let $\bar{\mu}^l = \left( \mu^l - \mu_G^l \right) \oslash \left( \mu^l - \mu_G^l \right)$ or $\bar{\mu}^l = \left( \tilde{\mu}^l - \tilde{\mu}_G^l \right) \oslash \left( \tilde{\mu}^l - \tilde{\mu}_G^l \right)$, depending on whether we measure collapse on $X$ or $\tilde{X}$, respectively.

## 4 Average gradient outer product induces DNC in Deep RFM

We demonstrate that AGOP is a mechanism for DNC in the Deep RFM model. In this section, we provide an extensive empirical demonstration that Deep RFM exhibits DNC - NC1 and NC2 - and that the induction of neural collapse is due to the application of AGOP.

We visualize the formation of DNC in the input vectors of each layer of Deep RFM. For $l \geqslant 2$, let $\tilde{X}_l \equiv M_{l-1}^{1/2} X_{l-1}$ be the feature vectors at layer $l - 1$ projected onto the square root of the AGOP, $M_{l-1}^{1/2}$, at that layer. Otherwise, we define $\tilde{X}_1 = X_1$ as the untransformed representations. In Figure 1, we show the Gram matrix, $\bar{X}_l^\top \bar{X}_l$, of centered and normalized feature vectors $\bar{X}_l = \left( \tilde{X}_l - \tilde{\mu}_G^l \right) \oslash \| \tilde{X}_l - \tilde{\mu}_G^l \|$ extracted from several layers of Deep RFM, as presented in Algorithm 1,

trained on CIFAR-10 and MNIST (see Appendix E for similar results on SVHN). Here the global mean $\widetilde{\mu}_G^l$ is subtracted from each column of $\widetilde{X}$. After 18 layers of DeepRFM, the final Gram matrix is approximately equal to that of collapsed data, in which all points are at exactly their class means, and the centered class means form an ETF. Specifically, all centered points of the same class have inner product 1, while pairs of different class have inner product $-(K-1)^{-1}$. Note that, like standard neural networks, Deep RFM exhibits DNC even when the classes are highly imbalanced, such as for the SVHN dataset (Figure 5B in Appendix E).

We show collapse occurring as a consequence of projection onto the AGOP, across all datasets and across choices of feature map (Figures 3 and 4 in Appendix E). We observe that the improvement in NC1 is entirely due to $M_l^{1/2}$, and the random feature map actually worsens the NC1 value. This result is intuitive as Deep RFM deviates from a simple deep random feature model just by additionally projecting onto the AGOP with respect to the input at each layer, and we do not expect random feature maps to induce neural collapse on their own. In fact, the random feature map will provably separate nearby datapoints, as this map is equivalent to applying a rapidly decaying non-linear function to the inner products between pairs of points (described formally in Appendix A).

## 5 Theoretical explanations for DNC in Deep RFM

We have established empirically in a range of settings that Deep RFM induces DNC through AGOP. We now prove that DNC in Deep RFM (1) occurs in an asymptotic setting, and (2) can be viewed as an implicit bias of RFM as a kernel learning algorithm.

### 5.1 Asymptotic analysis

Many works have derived that, under mild conditions on the data and kernel, non-linear kernel matrices of high dimensional data are well approximated by linear kernels with a non-linear perturbation [Karoui, 2010, Adlam et al., 2019, Hu and Lu, 2022]. In this section, we provide a proof that Deep RFM will induce DNC when the predictor kernels $\{k_l\}_l$ and random feature maps $\{\Phi_l\}_l$ satisfy this property. In particular, in the high-dimensional setting in these works, non-linear kernel matrices, written $k(X) \equiv k(X, X)$ for simplicity, are well approximated by

$$k(X) \approx \gamma \mathbf{1}\mathbf{1}^\top + X^\top X + \lambda I,$$

where $\lambda \geqslant 0$ is the *perturbation* parameter. Following Adlam and Pennington [2020], for additional simplicity we consider centered kernels, where $\gamma = 0$. We associate with Deep RFM two separate (unrestricted) centered kernels $\widehat{k}$ and $k_{\text{map}}$, with perturbation parameters $\widehat{\lambda}$ and $\lambda_{\text{map}}$, corresponding to the predictor kernels and random feature maps respectively.

We will show that Deep RFM exhibits exponential convergence to DNC, and the convergence rate depends on the ratio $\widehat{\lambda}/\lambda_{\text{map}}$. These two parameters modulate the distance of non-linear ridge regression with each kernel to linear regression, and therefore the extent of DNC with Deep RFM. Namely, as we will show, if $\widehat{k}$ is close to the linear kernel, i.e., if $\widehat{\lambda}$ is small, then the predictor in each layer resembles interpolating linear regression, an ideal setting for collapse through the AGOP. On the other hand, if $k_{\text{map}}$ is close to the linear kernel, i.e., if $\lambda_{\text{map}}$ is small, then the data is not *easily* linearly separable. In that case, the predictor will be significantly non-linear in order to interpolate the labels, deviating from the ideal setting. We proceed by explaining specifically where $\widehat{k}$ and $k_{\text{map}}$ appear in Deep RFM and why linear interpolation induces collapse. We then conclude with the statement of our main theorem and an explanation of its proof.

We describe the Deep RFM iteration we analyze here, which has a slight modification. Instead of normalizing the data at each iteration, we scale the data by a value $\kappa^{-1}$ at each iteration. This modification is sufficient to control the norms of the data, and is easier to analyze. The recursive procedure begins with a dataset $X_l$ at layer $l$ in Deep RFM, and constructs $\widetilde{X}_l$ from the AGOP $M_l$ of the kernel ridgeless regression solution with kernel $k_l = \widehat{k}$. After transforming with the AGOP and scaling, the data Gram matrix at layer $l$ is transformed to,

$$\widetilde{X}_{l+1}^\top \widetilde{X}_{l+1} = \kappa^{-1} X_l^\top M_l X_l, \tag{3}$$

for $\kappa = 1 - 2\widehat{\lambda}(1 + \lambda_{\mathrm{map}}^{-1})$. Then, we will apply a non-linear feature map $\Phi_{\mathrm{map}}$ corresponding to a kernel $k_{\mathrm{map}}$, and write the corresponding Gram matrix at depth $l + 1$ as

$$X_{l+1}^\top X_{l+1} = \Phi_{\mathrm{map}}(\tilde{X}_{l+1})^\top \Phi_{\mathrm{map}}(\tilde{X}_{l+1}) = k_{\mathrm{map}}(\tilde{X}_{l+1}) = \tilde{X}_{l+1}^\top \tilde{X}_{l+1} + \lambda_{\mathrm{map}} I \ . \quad (4)$$

Intuitively, the ratio $\widehat{\lambda}/\lambda_{\mathrm{map}}$ determines how close to linear $\widehat{k}$ is when applied to the dataset, and therefore the rate of NC formation.

We now explain why Deep RFM with a linear kernel is ideal for NC, inducing NC in just one AGOP application. To see this, note that ridgeless regression with a linear kernel is exactly least-squares regression on one-hot encoded labels. In the high-dimensional setting we consider, should we find a linear solution $f(x) = \beta^\top x$ that interpolates the labels, the AGOP of $f$ is $\beta\beta^\top$. Since we interpolate the data, $\beta^\top x = y$ for all $(x, y)$ input/label pairs, applying the AGOP collapses the data to $\beta\beta^\top x = \beta y$. Therefore, NC will occur in a single layer of Deep RFM when $\lambda_{\mathrm{map}} \gg \widehat{\lambda}$, in which case the data Gram matrix has a large minimum eigenvalue relative to the identity perturbation to $\widehat{k}$, so the kernel regression is effectively linear regression. Note that this theory offers an explanation why a non-linear activation is needed for DNC in neural networks: $\lambda_{\mathrm{map}} = 0$ when $\Phi_{\mathrm{map}}$ is linear, preventing this ideal setting.

We prove that DNC occurs in the following full-rank, high-dimensional setting.

**Assumption 5.1** (Data is high dimensional). We assume that the data has dimension $d \geqslant n$.

**Assumption 5.2** (Data is full rank). We assume that the initial data Gram matrix $X_1^\top X_1$ has minimum eigenvalue at least $\lambda_\phi > 0$.

The assumptions that the Gram matrix of the data is full-rank and high-dimensional is needed only if one requires neural collapse in every layer of Deep RFM, starting from the very first one. In contrast, if we consider collapse starting at any given later layer of Deep RFM, then we only need that the smallest eigenvalue of the corresponding feature map is bounded away from 0. This in turn requires that the number of features at that layer is greater than the number of data points, a condition which is routinely satisfied by the overparameterized neural networks used in practice.

We present our main theorem.

**Theorem 5.3** (Deep RFM exhibits neural collapse). *Suppose we apply Deep RFM on any dataset $X$ with labels $Y \in \mathbb{R}^{N \times K}$ choosing all $\{\Phi_l\}_l$ and $\{k_l\}_l$ as the feature map $\Phi_{\mathrm{map}}$ and kernel $\widehat{k}$ above, with no ridge parameter ($\gamma = 0$). Then, there exists a universal constant $C > 0$, such that for any $0 < \epsilon \leqslant 1$, provided $\widehat{\lambda} \leqslant \frac{C\lambda_{\mathrm{map}}}{2(1+\lambda_{\mathrm{map}}^{-1})n}(1-\epsilon)$, and for all layers $l \in \{2, \dots, L\}$ of Deep RFM,*

$$\|\tilde{X}_{l+1}^\top \tilde{X}_{l+1} - Y^\top Y\| \leqslant (1-\epsilon)\|\tilde{X}_l^\top \tilde{X}_l - Y^\top Y\| + O(\widehat{\lambda}^2 \lambda_{\mathrm{map}}^{-2}).$$

This theorem immediately implies exponential convergence to NC1 and NC2 up to error $O(L\widehat{\lambda}^2 \lambda_{\mathrm{map}}^{-2})$, a small value determined by the parameters of the problem. As a validation of our theory, we see that this exponential improvement in the DNC metric occurs in all layers (see Figures 3 and 4). Note the exponential rate predicted by Theorem 5.3 and observed empirically for Deep RFM is consistent with the exponential rate observed in deep neural networks [He and Su, 2022].

The proof for this theorem is roughly as follows. Recall that, by our argument earlier in this section, a linear kernel will cause collapse within just one layer of Deep RFM. However, in the more general case we consider, a small non-linear deviation from a linear kernel, $\widehat{\lambda}$, is introduced. Beginning with the first layer, partial collapse occurs if the ratio $\widehat{\lambda}/\lambda_{\mathrm{map}}^{-1}$ is sufficiently small. Following the partial collapse, in subsequent layers, the data Gram matrix will sufficiently resemble the collapsed matrix, so that the non-linear kernel solution on the collapsed data will behave like the linear solution, leading to further convergence to the collapsed data Gram matrix, a fixed point of the Deep RFM iteration (Lemma C.2).

## 5.2 Connection to parametrized kernel ridge regression

Next, we demonstrate that DNC emerges in parameterized kernel ridge regression (KRR). In fact, DNC arises as a natural consequence of minimizing the norm of the predictor jointly over the choice of kernel function and the regression coefficients. This result proves that DNC is an implicit bias of

kernel learning. We connect this result to Deep RFM by providing intuition that the kernel learned through RFM effectively minimizes the parametrized KRR objective and, as a consequence, RFM learns a kernel matrix that is biased towards the collapsed Gram matrix $Y^\top Y$.

We define the parametrized KRR problem. Consider positive semi-definite kernels (p.s.d.) $k_M : \mathbb{R}^d \times \mathbb{R}^d \to \mathbb{R}$ of form $k_M(x, z) = \phi(\|x - z\|_M)$, where $\|x - z\|_M = \sqrt{(x - z)^\top M(x - z)}$, $\phi$ is a strictly decreasing, strictly positive univariate function s.t. $\phi(0) = 1$, and $M$ is a p.s.d. matrix. This class of kernels covers a wide range of kernels including the Gaussian and Laplace we use in our experiments. Let $\mathcal{H}_M$ be the Reproducing Kernel Hilbert Space (RKHS) corresponding to our chosen kernel $k_M$ where functions $f \in \mathcal{H}_M$ map to a single output class. The parametrized kernel ridge regression for a single output class ($K = 1$) with ridge parameter $\gamma$ corresponds to the following optimization problem over such $f$ and $M$ on dataset $X$ and labels $Y \in \mathbb{R}^{K \times N}$:

$$\inf_{f,M} \frac{1}{2} \|f(X) - Y\|^2 + \frac{\gamma}{2} \|f\|^2_{\mathcal{H}_M} . \tag{5}$$

When we predict outputs over $K > 1$ classes, we jointly optimize over $K$ independent scalar-valued $f$, one for each output class, and, again, a single choice of matrix $M$. The objective function is then sum the individual objective values over all classes. Our modification for multiple outputs corresponds to how Deep RFM and kernel ridge regression, more generally, are solved in practice - a single kernel matrix is used across all classes while kernel coefficients are computed for each class independently. Note this modification for multiple classes is also equivalent to optimizing over a single vector-valued function where the function space $\mathcal{H}_M$ is a particular vector-valued RKHS (see Appendix B for more details).

Parameterized KRR differs from standard KRR by additionally optimizing over the choice of the kernel through the matrix $M$. For all $M$, including the optimal one for Problem (5), an analog of the representer theorem for vector-valued RKHS can be shown [Micchelli and Pontil, 2004] and the optimal solution to (5) can be written as $f(z) = \sum_{c,i=1,1}^{K,n} \alpha_{ci} k_M(x_{ci}, z)$, where $\alpha_{ci}$ is a $K$-dimensional vector. Let us therefore denote $A$ the matrix of stacked columns $\alpha_{ci}$. We can re-formulate Problem (5) as the following finite-dimensional optimization problem:

$$\inf_{A,M} \mathrm{tr}\left((Y - Ak_M)(Y - Ak_M)^T\right) + \mu \, \mathrm{tr}(k_M A^T A), \tag{6}$$

where, abusing notation, $k_M := k_M(X, X)$ is the matrix of pair-wise kernel evaluations on the data.

We now relax the optimization over matrices $M$ by optimizing over all p.s.d., entry-wise non-negative kernel matrices $k \in \mathbb{R}^{nK} \times \mathbb{R}^{nK}$ with ones on diagonal and compute the optimal value of the following relaxed parametrized KRR minimization:

$$\inf_{A,k} \mathrm{tr}\left((Y - Ak)(Y - Ak)^T\right) + \mu \, \mathrm{tr}(k A^T A). \tag{7}$$

Optimizing over all p.s.d., entry-wise non-negative $k$ with ones on diagonal is not always equivalent to optimizing over all $M$. Thus, in general, this provides only a relaxation. However, if the data $X$ is full column rank (as in our asymptotic analysis), then the optimizations (6) and (7) have the same solution and, thus, the relaxation is without loss of generality. We establish this equivalence formally in Appendix B.

We are now ready to state our optimality theorem. The proof is deferred to Appendix C.

**Theorem 5.4.** *The unique optimal solution to the (relaxed) parametrized kernel ridge regression objective (Problem 7) is the kernel matrix $k^*$ exhibiting neural collapse, $k^* = I_K \otimes (\mathbf{1}_n \mathbf{1}_n^\top) = Y^\top Y$.*

Finally, we informally connect this result to RFM. We argue that RFM naturally minimizes the parametrized kernel ridge regression objective (Problem 5) through learning the matrix $M$. This claim is natural since the RFM is a parametrized kernel regression model, where $M$ is set to be the AGOP. This choice of Mahalanobis matrix should minimize (5) as AGOP captures task-specific low-rank structure, reducing unnecessary variations of the predictor in irrelevant directions Chen et al. [2023]. Then, given any setting of $M$, RFM is conditionally optimal w.r.t. the parameter $\alpha$, as this method simply solves the original kernel regression problem for each fixed $M$. Further, as argued in Section 5.1, choosing $k$ to be a dot product kernel and $M$ to be the AGOP of an interpolating linear classifier implies that $k_M = k^*$, the optimal solution outlined in Theorem 5.4.

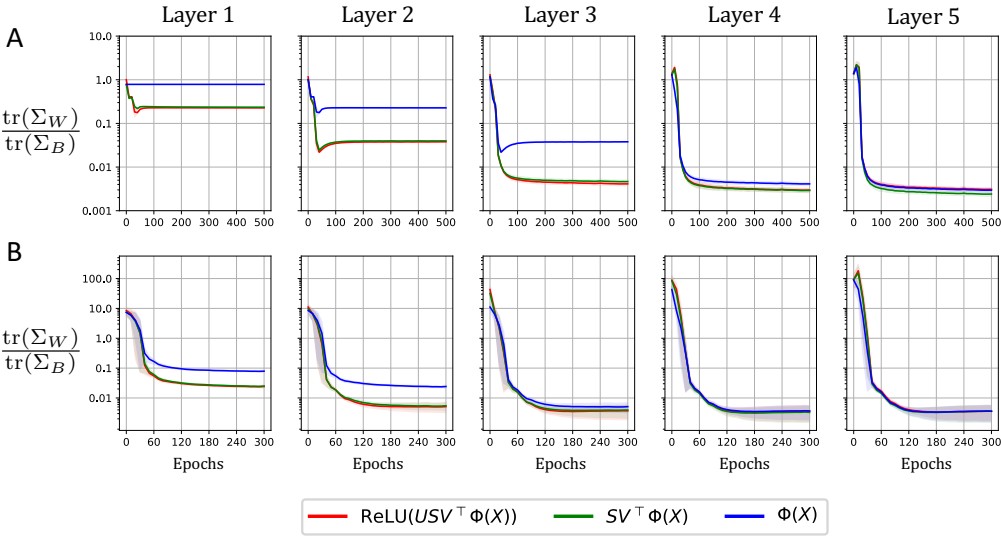

Figure 2: Feature variability collapse from different singular value decomposition components in (A) an MLP on MNIST, and (B) a ResNet on CIFAR-10. We measure the reduction in the NC1 metric throughout training at each of five fully-connected layers. Each layer is decomposed into its input, $\Phi(X)$, the projection onto the right singular space of $W$, $SV^\top\Phi(X)$, and then $U$, the left singular vectors of $W$, and the application of the non-linearity.

## 6 Within-class variability collapse through AGOP in neural networks

Thus far, we have demonstrated that Deep RFM exhibits DNC empirically, and have given theoretical explanations for this phenomenon. Here, we provide evidence that the DNC mechanism of Deep RFM, i.e., the projection onto the AGOP, is responsible for DNC formation in typical neural networks, such as MLPs, VGG, and ResNet trained by SGD with small initialization. We do so by demonstrating that within-class variability collapse occurs predominantly through the multiplication by the right singular structure of the weights. As the NFM, which is determined by the right singular structure of each weight matrix, is highly correlated with the AGOP, our results imply the AGOP is responsible for NC1 progression.

A fully-connected layer $l$ of a neural network consists of two components: multiplication by a weight matrix $W_l$, followed by the application of an element-wise non-linear activation function $\phi$ (termed the non-linearity). Both components of the layer crucially contribute to the inference and training processes of a neural network. The question we address in this section is whether the non-linearity or the weight matrix is primarily responsible for the improvement in DNC1 metrics.

We additionally decompose the weight matrix into its singular value decomposition $W_l = U_l S_l V_l^\top$, viewing a fully-connected layer as first applying $S_l V_l^\top$, then applying the composition of $\phi$ with $U_l$, $\phi(U_l\cdot)$. This decomposition allows us to directly consider the effect of the NFM, $W_l^\top W_l = V_l S_l^2 V_l^\top$, and therefore the AGOP, on DNC formation.

The grouping of a layer into the right singular structure and the non-linearity with the left singular vectors is natural, as DNC1 metrics computed on this decomposition are identical to the metrics computed using the whole $W_l$ and then the non-linearity. We see this fact by considering the standard DNC1 metric $\operatorname{tr}(\Sigma_W)\operatorname{tr}(\Sigma_B)^{-1}$ [Tirer et al., 2023, Rangamani et al., 2023]. A simple computation using the cyclic property of the trace gives $\operatorname{tr}(\hat\Sigma_W)\operatorname{tr}(\hat\Sigma_B)^{-1} = \operatorname{tr}(U\Sigma_W U^\top)\operatorname{tr}(U\Sigma_B U^\top)^{-1} = \operatorname{tr}(\Sigma_W)\operatorname{tr}(\Sigma_B)^{-1}$, where $\Sigma$ matrices refer to the within-class variability after the application of $SV^\top$, while the $\hat\Sigma$ matrices correspond to the output of the full weight matrix.

While both the non-linearity and $S_l V_l^\top$, and only these two components, can influence the DNC1 metric, we demonstrate that $S_l V_l^\top$ is responsible for directly inducing the majority of within-class variability collapse in neural networks trained by SGD with small initialization. We verify this claim

by plotting the DNC1 metrics of all layers of an MLP and ResNet network trained on MNIST and CIFAR-10, respectively (Figure 2), where each layer is decomposed into its different parts – (1) the input to that layer, (2) the embedding after multiplication with $S_l V_l^\top$, and (3) the embedding after multiplication with the left singular vectors $U_l$ and application of the non-linearity $\phi$.

We see that the ratio of within-class to between-class variability decreases mostly between steps (1) and (2), due to the application of the right singular structure. Similar results are in Appendix E for all combinations of datasets (MNIST, CIFAR-10, SVHN) and architectures (MLP, VGG, and ResNet). This is a profound insight into the underlying mechanisms for DNC that is of independent interest, especially given we train with a standard algorithm across many combinations of datasets and models.

We note that, while in this setting the ReLU does not directly reduce NC1, the ReLU is still crucial for ensuring the expressivity of the feature vectors. Without non-linearity, the neural network cannot interpolate the data or perform proper feature learning, necessary conditions for DNC to occur at all.

These results are in a regime where the NFA holds with high correlation, with ResNet and MLP having NFA correlations at the end of training (averaged over all layers and seeds) of $0.74 \pm 0.17$ and $0.74 \pm 0.13$ on CIFAR-10 and MNIST, respectively (see Appendix E for values across all architectures and datasets). Therefore, as $W_l^\top W_l = V_l S_l^2 V_l^\top$, $S_l V_l^\top$ should project into a similar space as the AGOP. We note that the matrix quantities involved are high-dimensional, and the trivial correlation between two i.i.d. uniform eigenvectors of dimension $d$ concentrates within $\pm O(d^{-1/2})$. For the width $512$ weight matrices considered here, this correlation would be approximately (at most) $0.04$. Therefore, we conclude that the AGOP structure can decrease within-class variability in DNNs.

Unlike DNC1, we have observed a strong DNC2 progression only in the very last layer. This is due to the fact that DNC2 was observed in the related work Rangamani et al. [2023], Parker et al. [2023] in a regime of large initialization. In contrast, the NFA holds most consistently with small initialization [Radhakrishnan et al., 2024a, Beaglehole et al., 2024], and in this feature learning regime, the low-rank bias prevents DNC2 [Li et al., 2020, Súkeník et al., 2024].

## 7 Conclusion

This work establishes that deep neural collapse can occur through feature learning with the AGOP. We bridge the unsatisfactory gap between DNC and the data – with previous work mostly only focusing on the very end of the network and ignoring the training data.

We demonstrate that projection onto the AGOP induces deep neural collapse in Deep RFM. We validate that AGOP induces NC in Deep RFM both empirically and theoretically through asymptotic and kernel learning analyses.

We then provide evidence that the AGOP mechanism of Deep RFM induces DNC in general neural networks. We experimentally show that the DNC1 metric progression through the layers can be mostly due to the linear denoising via the application of the center-right singular structure. Through the NFA, the application of this structure is approximately equivalent to projection onto the AGOP, suggesting that the AGOP directly induces within-class variability collapse in DNNs.

## Acknowledgements

We acknowledge support from the National Science Foundation (NSF) and the Simons Foundation for the Collaboration on the Theoretical Foundations of Deep Learning through awards DMS-2031883 and #814639 as well as the TILOS institute (NSF CCF-2112665). This work used the programs (1) XSEDE (Extreme science and engineering discovery environment) which is supported by NSF grant numbers ACI-1548562, and (2) ACCESS (Advanced cyberinfrastructure coordination ecosystem: services & support) which is supported by NSF grants numbers #2138259, #2138286, #2138307, #2137603, and #2138296. Specifically, we used the resources from SDSC Expanse GPU compute nodes, and NCSA Delta system, via allocations TG-CIS220009. Marco Mondelli is supported by the 2019 Lopez-Loreta prize. We also acknowledge useful feedback from anonymous reviewers.

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

# A    Random features cannot significantly reduce within-class variability

We state the following proposition (first appearing in Cho and Saul [2009]) that random features mapping with ReLU will separate distinct data points, especially nearby ones. Therefore, we expect theoretically that NC1 metrics will not improve due to this random feature map.

**Proposition A.1.** *[Cho and Saul [2009]] Let $x, y \in \mathbb{R}^d$ be two fixed vectors of unit length. Let $W \in \mathbb{R}^{D \times d}$ be a weight matrix whose entries are initialized i.i.d. from $\mathcal{N}(0, 2/D)$. Assume $x^\top y = r$. Then, for $\sigma(\cdot) = \text{ReLU}(\cdot)$,*

$$\mathbb{E}(\sigma(Wx)^\top \sigma(Wy)) = \frac{1}{\pi} \left( \sin \theta + (\pi - \theta) \cos \theta \right) , \tag{8}$$

*where $\theta = \cos^{-1} r$. Moreover, the variance of this dot product scales with $1/D$, and the product is sub-exponential, making it well-concentrated.*

This proposition gives a relationship between the dot product $x^\top y$ of two vectors and the dot product of the outputs of their corresponding random feature maps $\sigma(Wx)^\top \sigma(Wy)$. This is relevant because for unit vectors, the dot product is a direct indication of the distance between the two vectors.

This means that the distances between data points generally increase, but not drastically. They increase irrespective of the angle between data points, however they tend to expand relatively more, if the angle $\theta$ is close to $0$. Therefore, the DNC1 metric should on average marginally increase or stay constant. This is in agreement with our Deep RFM measurements on all datasets we consider (Appendix E), for both ReLU and cosine activations. Importantly, this also supports the connection between neural networks and Deep RFM, since in DNNs the DNC1 metric did not significantly change after applying the ReLU in our setting (e.g. Figure 2).

# B    Additional results on parametrized kernel ridge regression

**Vector-valued Reproducing Kernel Hilbert Spaces**    Since we consider multi-class classification, the function we learn is multi-output and therefore we have to use vector-valued kernel ridge regression in the most general case. Similar to kernel ridge regression with a single output, its multi-dimensional generalization also has interpretation through minimization in an RKHS function space, just with functions of a vector-valued RKHS. The vector-valued RKHS is a Hilbert space $\mathcal{H}$ of vector-valued functions on a domain $\mathcal{X}$ such that there exists a positive-definite matrix-valued function $\Gamma : \mathcal{X} \times \mathcal{X} \to \mathbb{R}^{K \times K}$ such that for any $x \in \mathcal{X}$ and any probe vector $c \in \mathbb{R}^{K \times 1}$, the function $\Gamma(x, \cdot)c$ is in $\mathcal{H}$ and a reproducing property similar to scalar valued RKHS holds: $f(x)^\top c = \langle f, \Gamma(x, \cdot)c \rangle_{\mathcal{H}}$. For more on vector-valued RKHS see e.g. Ciliberto et al. [2015].

In our setting, we only consider a subclass of vector-valued RKHS in which the matrix valued function $\Gamma$ can be decomposed as $\Gamma(x, z) = k(x, z) \cdot I_K$ for some scalar-valued kernel $k : \mathcal{X} \times \mathcal{X} \to \mathbb{R}$. This particular type of RKHS is referred to as separable and is the parametrization used for Deep RFM and kernel ridge regression in practice.

Now we formally connect the Problems 6 and 7 under the assumption that the data $X$ has full column rank.

**Proposition B.1.** *When the data Gram matrix $X$ has minimum eigenvalue $\lambda > 0$ (otherwise assuming the same setup as Section 5.2), the relaxation in Problem (7) has the same solution as Problem (6).*

*Proof of Proposition B.1.* We first show any $k$ realizable (as $\phi$ applied to the Euclidean distance on some dataset $R$) can be constructed by applying $k_M$ to our data $X$ under Mahalanobis distance with appropriately chosen $M$ matrix. Let $k$ be a realizable matrix - i.e. any desired positive semi-definite kernel matrix for which there exists a dataset $R'$ that satisfies $k = \phi(d(R', R'))$, where $d : \mathbb{R}^{n \times d} \times \mathbb{R}^{n \times d} \to \mathbb{R}^{n \times n}$ denotes the matrix of Euclidean distances between all pairs of points in the first and second argument. Our construction first takes the entry-wise inverse $r = \phi^{-1}(k)$. Since $k$ is realizable, this $r$ must be a matrix of Euclidean distances between columns of a data matrix $R$ of the same dimensions as $X$. Assuming the gram matrix of $X$ is invertible, we simply solve the system $R = NX$ for a matrix $N$ and set $M = N^\top N$ which yields $k = \phi(r) = \phi(d(R, R)) = \phi(d_M(X, X))$ where $d_M(\cdot, \cdot)$ denotes the operation that produces a Mahalanobis distance matrix for the Mahalanobis matrix $M$.

We now give a construction demonstrating that the solution $k^*$ to Problem (7) is realizable up to arbitrarily small error using our parametrized kernel on a dataset $R$ under Euclidean distance. We can realize the ideal kernel matrix that solves (6), $Y^\top Y$, up to arbitrarily small error by choosing $R$ according to a parameter $\epsilon > 0$, such that $\phi(d(R, R)) \to Y^\top Y$ as $\epsilon \to 0$. In particular, for feature vectors $x_i, x_j$ of the same label in columns $i, j$ of $X$, we set the feature vectors $R_i, R_j$ for columns $i, j$ in $R$ to have $\|R_i - R_j\| = 0$. Then, for $x_i, x_j$ in $X$ of different class, we set $R_i, R_j$ as columns of $R$ such that $\|R_i - R_j\| > \epsilon^{-1}$. Then $k(R_i, R_j)$ is identically 1 for $R_i, R_j$ from the same class and converges to 0 for points from different classes, as $\epsilon \to 0$, giving that $k = \phi(r) \to Y^\top Y$.

For any choice of $\epsilon > 0$, we can apply the procedure described two paragraphs above to construct $M$ that realizes the same $k$ as applying $k_M$ to our dataset under the Mahalanobis distance. Therefore, the solution to (7) can be constructed as the infimum over feasible solutions to (6), completing the proof. $\qquad\square$

# C  Additional proofs

## C.1  Asymptotic results

**Lemma C.1** (Woodbury Inverse Formula [Woodbury, 1950])**.**

$$\left(P + UV^\top\right)^{-1} = P^{-1} - P^{-1}U\left(I + V^\top P^{-1}U\right)^{-1}V^\top P^{-1}.$$

**Lemma C.2** (Fixed point of collapse)**.** *Let $A^* = Y^\top Y + \lambda_{\mathrm{map}}I$, the collapsed data gram matrix following an application of the non-linear random feature map $\Phi_{\mathrm{map}}$. Then,*

$$(A^*)^{-1} = \lambda_{\mathrm{map}}^{-1}I - \lambda_{\mathrm{map}}^{-1}\left(\lambda_{\mathrm{map}} + n\right)^{-1}Y^\top Y.$$

*Proof of Lemma C.2.* We write $Y^\top Y = nUU^\top$, where $U \in \mathbb{R}^{n \times K}$ is the matrix of normalized eigenvectors of $Y^\top Y$. By Lemma C.1 and that $U^\top U = I$,

$$
\begin{aligned}
(A^*)^{-1} &= \left(\lambda_{\mathrm{map}}I + nUU^\top\right)^{-1} \\
&= \lambda_{\mathrm{map}}^{-1}I - \frac{n}{\lambda_{\mathrm{map}}^2}U\left(I + n\lambda_{\mathrm{map}}^{-1}U^\top U\right)^{-1}U^\top \\
&= \lambda_{\mathrm{map}}^{-1}I - \frac{n}{\lambda_{\mathrm{map}}^2} \cdot \frac{1}{1 + n\lambda_{\mathrm{map}}^{-1}}UU^\top \\
&= \lambda_{\mathrm{map}}^{-1}I - \frac{n}{\lambda_{\mathrm{map}}(\lambda_{\mathrm{map}} + n)}UU^\top \\
&= \lambda_{\mathrm{map}}^{-1}I - \frac{\lambda_{\mathrm{map}}^{-1}}{\lambda_{\mathrm{map}} + n}Y^\top Y.
\end{aligned}
$$

$\qquad\square$

*Proof of Proposition 5.3.* We have solved the kernel ridgeless regression problem to get coefficients $\alpha = \left(X_l X_l^\top + \widehat{\lambda}I\right)^{-1}y$. Then, the predictor at layer $l$ evaluated on the training data is,

$$f_l(X_l) = (X_l^\top X_l + \widehat{\lambda}I)\alpha .$$

As in Deep RFM, let $M$ be the AGOP. Then,

$$M = \sum_{i=1}^{n} \nabla f(x^{(i)})(\nabla f(x^{(i)}))^\top = X_l \alpha \alpha^\top X_l^\top .$$

We drop the subscript $l$ for simplicity. Therefore,

$$X^\top M X = X^\top X \left(X^\top X + \widehat{\lambda}I\right)^{-1} Y^\top Y \left(X^\top X + \widehat{\lambda}I\right)^{-1} X^\top X .$$

Let $A = X^\top X$. Applying Lemma C.1,

$$\left(A + \widehat{\lambda}I\right)^{-1} = A^{-1} - \widehat{\lambda}A^{-1}\left(I + \widehat{\lambda}A^{-1}\right)^{-1}A^{-1} .$$

Therefore,

$$X^\top MX = \left(I - \widehat{\lambda}\left(I + \widehat{\lambda}A^{-1}\right)^{-1}A^{-1}\right)Y^\top Y\left(I - \widehat{\lambda}\left(I + \widehat{\lambda}A^{-1}\right)^{-1}A^{-1}\right) .$$

Applying that, by assumption, $\widehat{\lambda}\lambda_{\mathrm{map}}^{-1}$ is a small value $\epsilon_{\mathrm{lin}}$, and as $\lambda_{\mathrm{map}}$ is the minimum eigenvalue of $A$, we have that $\widehat{\lambda}\|A^{-1}\| \leqslant \widehat{\lambda}\lambda_{\mathrm{map}}^{-1} \triangleq \epsilon_{\mathrm{lin}}$. Therefore, $\left(I + \widehat{\lambda}A^{-1}\right)^{-1}\widehat{\lambda}A^{-1} = \widehat{\lambda}A^{-1} - \left(\widehat{\lambda}A^{-1}\right)^2 +$ $\cdots = \widehat{\lambda}A^{-1} + O(\epsilon_{\mathrm{lin}}^2)$, where $O(\epsilon_{\mathrm{lin}}^2)$ refers to some matrix of spectral norm at most $C\epsilon_{\mathrm{lin}}^2$ for some constant $C > 0$. Then,

$$X^\top MX = \left(I - \widehat{\lambda}A^{-1} + O(\epsilon_{\mathrm{lin}}^2)\right)Y^\top Y\left(I - \widehat{\lambda}A^{-1} + O(\epsilon_{\mathrm{lin}}^2)\right) = Y^\top Y - \widehat{\lambda}A^{-1}Y^\top Y - \widehat{\lambda}Y^\top Y A^{-1} + O(\epsilon_{\mathrm{lin}}^2) .$$

Let $\tilde{A} = \frac{A - A^*}{\|A - A^*\|}$, and $\epsilon_A = \|A - A^*\|$. Then, applying Lemma C.1,

$$A^{-1} = \left(A^* + \epsilon_A\tilde{A}\right)^{-1} = \left(I + \epsilon_A\left(A^*\right)^{-1}\tilde{A}\right)^{-1}\left(A^*\right)^{-1} = \left(A^*\right)^{-1} - \epsilon_A\left(A^*\right)^{-1}\left(I + \epsilon_A\tilde{A}\left(A^*\right)^{-1}\right)^{-1}\tilde{A}\left(A^*\right)^{-1} .$$

Let $\tilde{\Psi} = \left(I + \epsilon_A\tilde{A}\left(A^*\right)^{-1}\right)^{-1}$. Assuming partial collapse has already occurred, i.e., $\epsilon_A\lambda_{\mathrm{map}}^{-1} < 1/2$,

$$\|\tilde{\Psi}\| \leqslant \left(1 - \epsilon_A\lambda_{\mathrm{map}}^{-1}\right)^{-1} \leqslant 2 .$$

In this case, using that $Y^\top Y\left(A^*\right)^{-1} = \left(A^*\right)^{-1}Y^\top Y = (1 + \lambda_{\mathrm{map}}^{-1})Y^\top Y$ (Lemma C.2),

$$\begin{aligned}
X^\top MX &= Y^\top Y - \widehat{\lambda}A^{-1}Y^\top Y - \widehat{\lambda}Y^\top Y A^{-1} + O(\epsilon_{\mathrm{lin}}^2) \\
&= Y^\top Y - \widehat{\lambda}\left(A^*\right)^{-1}Y^\top Y - \widehat{\lambda}Y^\top Y\left(A^*\right)^{-1} + O(\epsilon_{\mathrm{lin}}^2) \\
&\quad - \widehat{\lambda}\epsilon_A\left(A^*\right)^{-1}\tilde{\Psi}\tilde{A}\left(A^*\right)^{-1}Y^\top Y - \widehat{\lambda}\epsilon_A Y^\top Y\left(A^*\right)^{-1}\tilde{A}\tilde{\Psi}\left(A^*\right)^{-1} \\
&= Y^\top Y - 2\widehat{\lambda}(1 + \lambda_{\mathrm{map}}^{-1})Y^\top Y + O(\epsilon_{\mathrm{lin}}^2) \\
&\quad - \widehat{\lambda}\epsilon_A(1 + \lambda_{\mathrm{map}}^{-1})\left(A^*\right)^{-1}\tilde{\Psi}\tilde{A}Y^\top Y - \widehat{\lambda}\epsilon_A(1 + \lambda_{\mathrm{map}}^{-1})Y^\top Y\tilde{A}\tilde{\Psi}\left(A^*\right)^{-1} .
\end{aligned}$$

Therefore, where $\kappa = 1 - 2\widehat{\lambda}(1 + \lambda_{\mathrm{map}}^{-1}) > 1/2$ by choice of $\widehat{\lambda} \cdot 2\lambda_{\mathrm{map}}^{-1}(1 + \lambda_{\mathrm{map}}^{-1})n < 1 - \epsilon$,

$$\begin{aligned}
\|X_{l+1}^\top X_{l+1} - A^*\| &= \|\kappa^{-1}X^\top MX - Y^\top Y\| \\
&= \left\|O(\epsilon_{\mathrm{lin}}^2) + \epsilon_A \cdot \widehat{\lambda} \cdot 2\kappa^{-1}(1 + \lambda_{\mathrm{map}}^{-1})\left(A^*\right)^{-1}\tilde{\Psi}\tilde{A}Y^\top Y\right\| \\
&\leqslant O(\epsilon_{\mathrm{lin}}^2) + \epsilon_A \cdot \widehat{\lambda} \cdot 2\lambda_{\mathrm{map}}^{-1}(1 + \lambda_{\mathrm{map}}^{-1})n \\
&< O(\epsilon_{\mathrm{lin}}^2) + \epsilon_A(1 - \epsilon) \\
&= O(\epsilon_{\mathrm{lin}}^2) + (1 - \epsilon)\|X_l X_l^\top - A^*\| .
\end{aligned}$$

It remains to show that partial collapse, $\epsilon_A\lambda_{\mathrm{map}}^{-1} < 1/2$, happens in the first iteration. To ensure this condition, recall,

$$X^\top MX = Y^\top Y - \widehat{\lambda}A^{-1}Y^\top Y - \widehat{\lambda}Y^\top Y A^{-1} + O(\epsilon_{\mathrm{lin}}^2) .$$

Therefore, because $\|A^{-1}\| \leqslant \lambda_{\mathrm{map}}^{-1}$, where $\Psi$ is some error matrix with norm 1,

$$X^\top MX = Y^\top Y + \widehat{\lambda}\lambda_{\mathrm{map}}^{-1}n\Psi + O(\widehat{\lambda}^2\lambda_{\mathrm{map}}^{-2}) .$$

Therefore,

$$\|X^\top MX - Y^\top Y\| < 1/2 ,$$

provided $\widehat{\lambda}\lambda_{\mathrm{map}}^{-1}n < C_1/2$, for some universal constant $C_1$. $\qquad\square$

## C.2 Non-asymptotic results

**Theorem.** *The unique optimal solution to the (relaxed) parametrized kernel ridge regression objective (Problem 7) is the kernel matrix $k^*$ exhibiting neural collapse, $k^* = I_K \otimes (\mathbf{1}_n \mathbf{1}_n^\top) = Y^\top Y$.*

*Proof of Theorem 5.4.* Denote $\mathcal{L}(A, k) = \operatorname{tr}\left(A(k^2 + \mu k)A^T\right) - 2\operatorname{tr}(AkY^T)$. Note that this problem is separable in the rows of $A$. If we only look at one row of $A$ at a time, we can solve this problem explicitly for that row if we assume that $k$ is fixed and positive definite, simply by solving solving for the zero gradient. Doing this for every row of $A$ and summing up, we get that Problem 6 can be reduced to the following problem:

$$\sup_k \sum_{c=1}^{K} Y_c^T (k + \mu I)^{-1} k Y_c,$$

where $Y_c \in \mathbb{R}^{N \times 1}$ is the vector of labels for class $c$. After writing the eigenvalue decomposition of $k$ as $VSV^T$ and using $(k + \mu I)^{-1} k = V(S + \mu I)^{-1} V^T V S V^T$, this supremum is equivalent to:

$$\sup_k \sum_{c=1}^{K} \sum_{i=1}^{Kn} (Y_c^T v_i)^2 \frac{\lambda_i}{\lambda_i + \mu},$$

where $\lambda_i, v_i$ are the $i$-th eigenvalue and eigenvector, respectively. By continuity of the Problem 6 in both variables, we can without loss of generality plug $k^*$ into this despite being low-rank. This can be seen by contradiction – if the $\mathcal{L}(A^*, k^*)$ would not be equal to $\sum_{c=1}^{K} Y_c^T (k^* + \mu I)^{-1} k^* Y_c$, we can find a converging sequence $k_i$ of symmetric PD kernel matrices that converges to $k^*$ for which the two objectives are equal and by continuity they converge to the objectives evaluated at $A^*, k^*$, thus they must be equal as well. After a simple computation we get:

$$\mathcal{L}(A^*, k^*) = Kn \frac{n}{n + \mu}.$$

With slight abuse of notation let us now re-scale $y$, so that each row is unit norm and divide the loss $\mathcal{L}$ by $K$. Then we get (abusing the notation):

$$\mathcal{L}(A^*, k^*) = \frac{n}{n + \mu}.$$

Now, $\sum_{i=1}^{Kn} (y^T v_i)^2 = 1$ since $(v_i)_{i=1}^{Kn}$ forms an orthogonal basis and therefore a tight frame with constant 1. Therefore the expression

$$\sup_k \frac{1}{K} \sum_{c=1}^{K} \sum_{i=1}^{Kn} (Y_c^T v_i)^2 \frac{\lambda_i}{\lambda_i + \mu}$$

can be viewed as a weighted average of the $\frac{\lambda_i}{\lambda_i + \mu}$ terms. To simplify the expression, denote $\omega_i := \frac{1}{K} \sum_{c=1}^{K} (Y_c^T v_i)^2$. Using Cauchy-Schwarz inequality on each individual summand, we see that $(Y_c^T v_i)^2 \leqslant \sum_{j; y_{cj}=1} v_{ij}^2$ and the equality is only achieved if all the entries of $v_i$ corresponding to entries of $Y_c$ equal to 1 are the same. Thus, we get $\sum_{c=1}^{K} (Y_c^T v_i)^2 \leqslant \|v_i\|^2 = 1$ with the equality if and only if, for each $c$, all the entries of $q_i$ corresponding to the entries of $Y_c$ equal to 1 are the same. This gives that $\omega_i \leqslant \frac{1}{K}$.

Now, take any feasible $k$. Then,

$$\mathcal{L}(k, \alpha^*) = \sum_{i=1}^{Kn} \omega_i \frac{\lambda_i}{\lambda_i + \mu} \leqslant \frac{1}{K} \sum_{i=1}^{K} \frac{\lambda_i}{\lambda_i + \mu} < \frac{\frac{1}{K} \sum_{i=1}^{K} \lambda_i}{\frac{1}{K} \sum_{i=1}^{K} \lambda_i + \mu} \leqslant \frac{n}{n + \mu}.$$

The first inequality is due to the monotonicity of the function $g(x) = \frac{x}{x + \mu}$ and $\omega_i \leqslant \frac{1}{K}$; the second strict inequality is due to Jensen after noting that $g$ is strictly convex and using the Perron-Frobenius theorem which says that $\lambda_1$ has multiplicity 1; and the last inequality is due to $\sum_{i=1}^{Kn} \lambda_i = Kn$, which follows from the well-known trace inequality and the fact that $k$ must have all the diagonal elements equal 1. This concludes the proof. $\qquad\square$

# D  Experimental details

In both Deep RFM and neural networks, we one hot encode labels and use $\pm 1$ for within/outside of each class.

For the neural network experiments, we use 5 hidden layer MLP networks with no biases and ReLU activation function. All experiments use SGD with batch size 128. We use default initialization for the linear readout layer. The layers all use width $512$ in all experiments. All models are trained with MSE loss. We measure the AGOP and NC metrics every 10 epochs. VGG was trained for 600 epochs with 0.9 momentum, learning rate 0.01, and initialization 0.3 in the intermediate layers (0.2 for MNIST). ResNet was trained for 300 epochs, no momentum, 0.2 init. scale in the intermediate layers, and learning rate 0.05. MLPs were trained for 500 epochs, no momentum, 0.3 init. scale (0.2 for MNIST), and learning rate 0.05.

We use the standard ResNet18 architecture (from the Pytorch `torchvision` library), where we replace the classifier (a fully-connected network) at the end with 5 MLP layers (each layer containing a linear layer followed by a ReLU, without any biases). We truncate the VGG-11, as defined in `torchvision.models`, to the ReLU just before the final two pooling layers, so that the pooling sizes matches MNIST, which contains 28x28 images, then attach at the end a 5-layer MLP as the classifier.

For the Deep RFM experiments, we generally use the Laplace kernel $k$, which evaluates two datapoints $x, z \in \mathbb{R}^d$ as $k(x, z) = \exp\left(-\|x - z\|_2/L\right)$, where $L$ is a bandwidth parameter. For the experiments with ReLU on MNIST, we use the Gaussian kernel with bandwidth $L$ instead. In these experiments, we set $L = 2.0$. We perform ridgeless regression to interpolate the data, i.e. $\mu = 0$. We use width 1024 for the random feature map in Deep RFM with ReLU activation function. For experiments with the cosine activation, we used Random Fourier Features corresponding to the $\ell_1$-Laplacian kernel with bandwidth $\sigma = 0.05$ and 4096 total features.

Unless otherwise specified, we perform experiments using the first 50,000 points from each of MNIST, SVHN, and CIFAR-10 loaded with `torchvision` datasets.

For neural network experiments, we averaged over 3 seeds and report 1 standard deviation error bars, truncating the lower interval to $0.01\times$ the mean value. We report Pearson correlation for the NFA values (where each matrix is subtracted by its mean value). As noted in the main text, we compute correlation of the NFM with the square root of the AGOP as in Radhakrishnan et al. [2024a].

Each experiment was performed on a single NVIDIA A100 GPU. Each experiment was completed in under 1.5 GPU hours.

All code is available at this link: `https://github.com/dmbeaglehole/neural_collapse_rfm`. See in particular `nc_nn.py` and `deep_rfm.py` for NN and Deep RFM code.

The code for Random Fourier Features (used in Deep RFM with cosine activation function) was adapted from `https://github.com/hichamjanati/srf.git`.

# E  Additional plots

In this section, we give the full set of results for all combinations of datasets and neural network architectures for Deep RFM and standard DNNs (VGG, ResNet, and MLPs). In Figures 3 and 5, we plot the NC1 and NC2 metrics for Deep RFM as a function of the layer for ReLU and cos activation functions. In Figures 4 and 6, we visualize the formation of DNC in Deep RFM as in the main text. In Figures 7, 8, and 9, we demonstrate that the right singular structure can reduce the majority of NC1 reduction in MLPs, ResNet, and VGG, respectively. In Figures 10, 11, and 12, we verify the NFA, DNC, and plot the training loss.

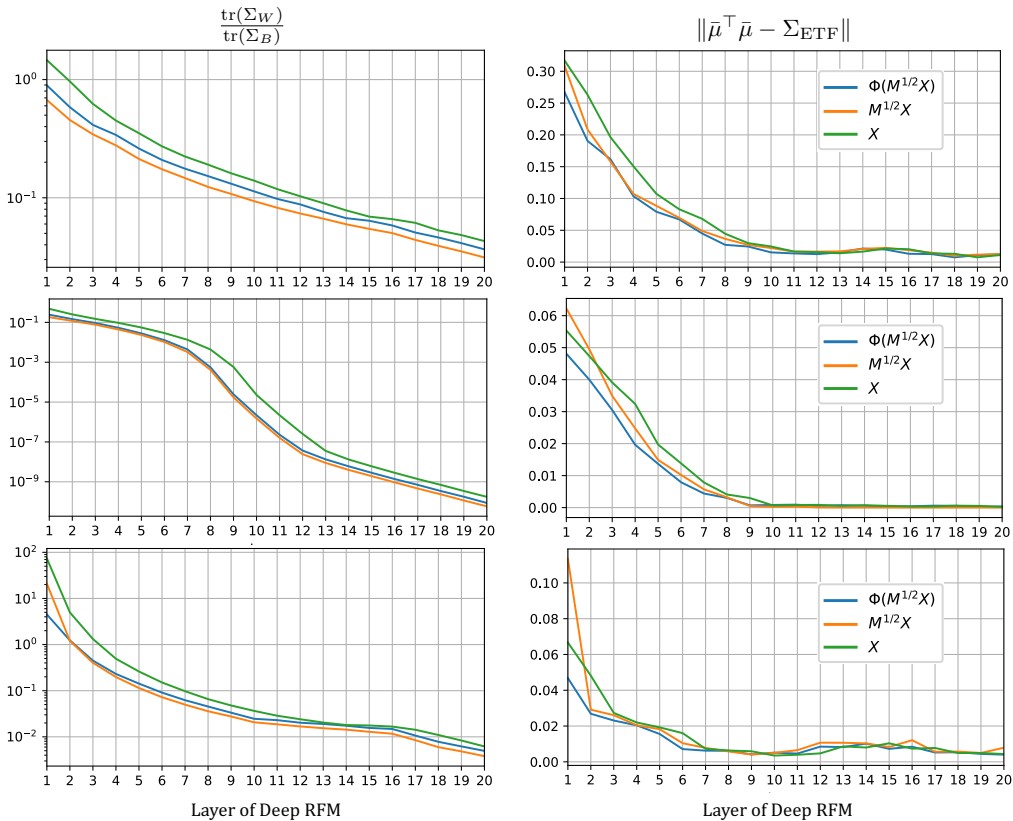

Figure 3: Neural collapse with Deep RFM on additional datasets with $\sigma(\cdot) = \mathrm{ReLU}(\cdot)$. We show $\mathrm{tr}\,\Sigma_W/\mathrm{tr}\,\Sigma_B$, our NC1 metric on the left, and $\left\|\tilde{\mu}\tilde{\mu}^\top - \Sigma_{\mathrm{ETF}}\right\|$, our NC2 metric, on the right. The first row is CIFAR-10, second is MNIST, third is SVHN. We plot these metrics as a function of depth of Deep RFM for the original data $X$ (green), the data after applying the square root of the AGOP $M_l^{1/2}x$ (orange), and the data after the AGOP and non-linearity (blue).

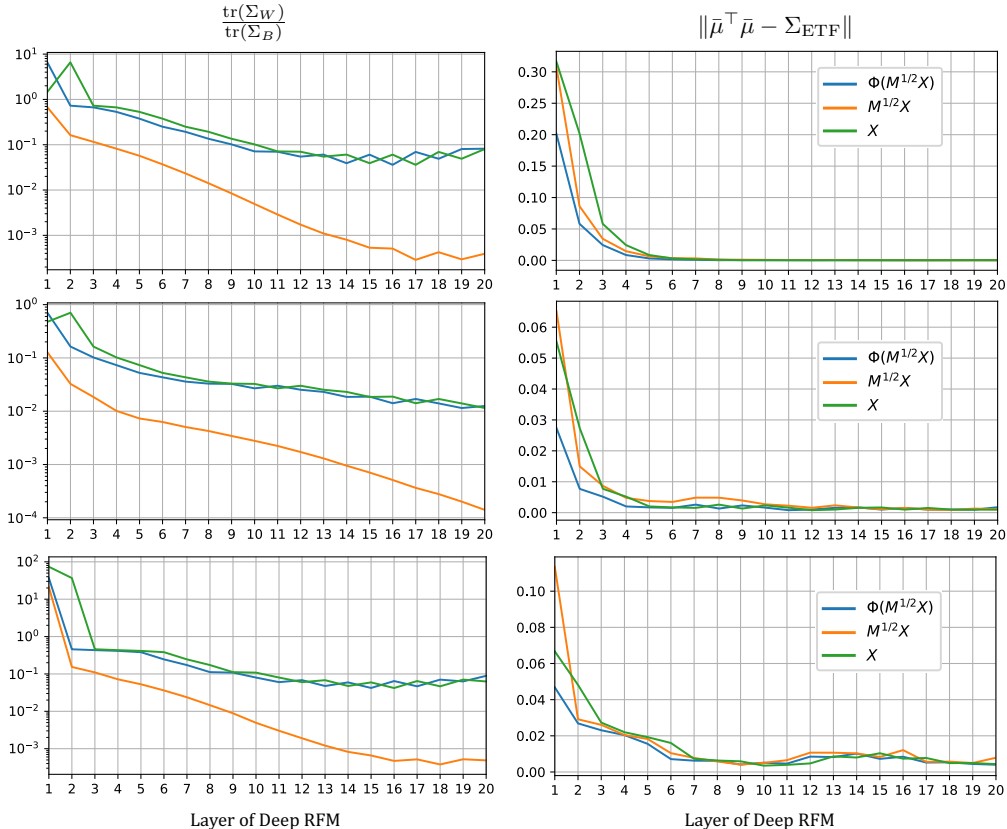

Figure 4: Neural collapse with Deep RFM on additional datasets with $\sigma(\cdot) = \cos(\cdot)$. We show $\operatorname{tr}\Sigma_W / \operatorname{tr}\Sigma_B$, our NC1 metric on the left, and $\left\|\tilde{\mu}\tilde{\mu}^\top - \Sigma_{\mathrm{ETF}}\right\|$, our NC2 metric, on the right. he first row is CIFAR-10, second is MNIST, third is SVHN. We plot these metrics as a function of depth of Deep RFM for the original data $X$ (green), the data after applying the square root of the AGOP $M_l^{1/2}x$ (orange), and the data after the AGOP and non-linearity (blue).

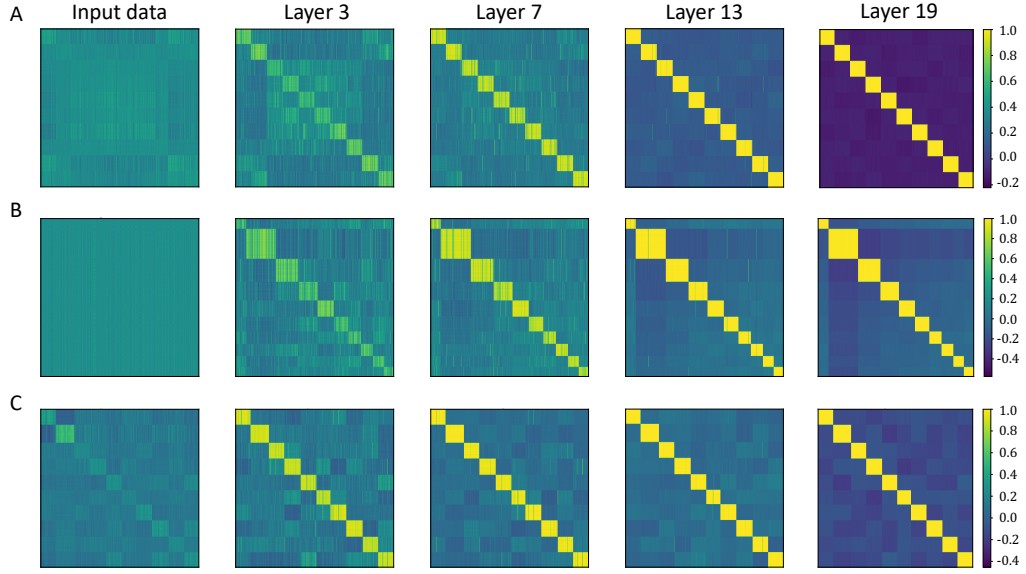

Figure 5: Visualization of neural collapse for Deep RFM on additional datasets with $\sigma(\cdot) = \cos(\cdot)$. As in the main text, we plot the Gram matrix of the centered and normalized feature vectors $\widetilde{X}_l$. We see the data form the ETF in the final column. (A) corresponds to CIFAR-10, (B) SVHN, and (C) MNIST.

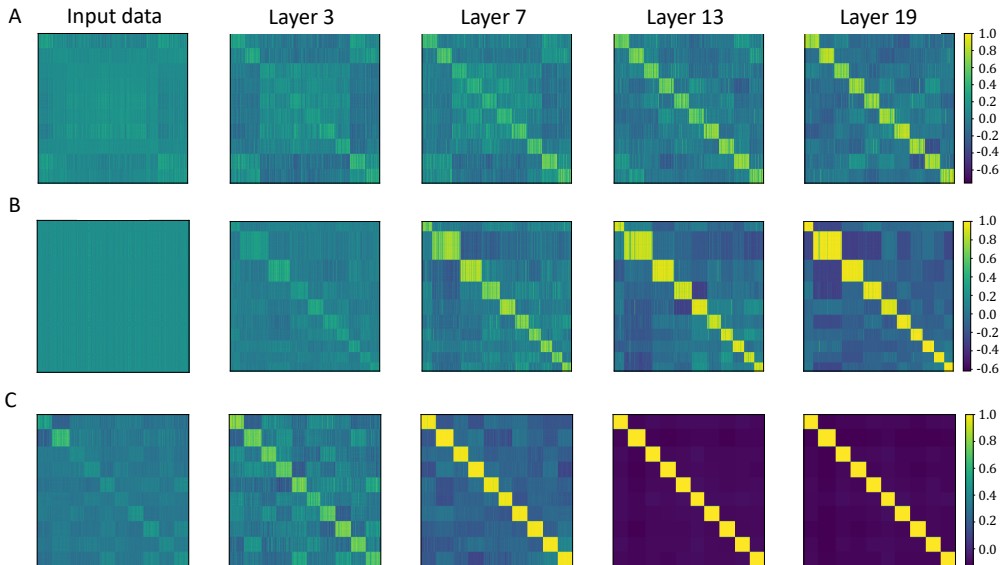

Figure 6: Visualization of neural collapse for Deep RFM on additional datasets with $\sigma(\cdot) = \mathrm{ReLU}(\cdot)$. As in the main text, we plot the Gram matrix of the centered and normalized feature vectors $\widetilde{X}_l$. The first column displays the Gram matrix of the untransformed (but normalized and centered) data. We see the data form the ETF in the final column. (A) corresponds to CIFAR-10, (B) SVHN, and (C) MNIST.

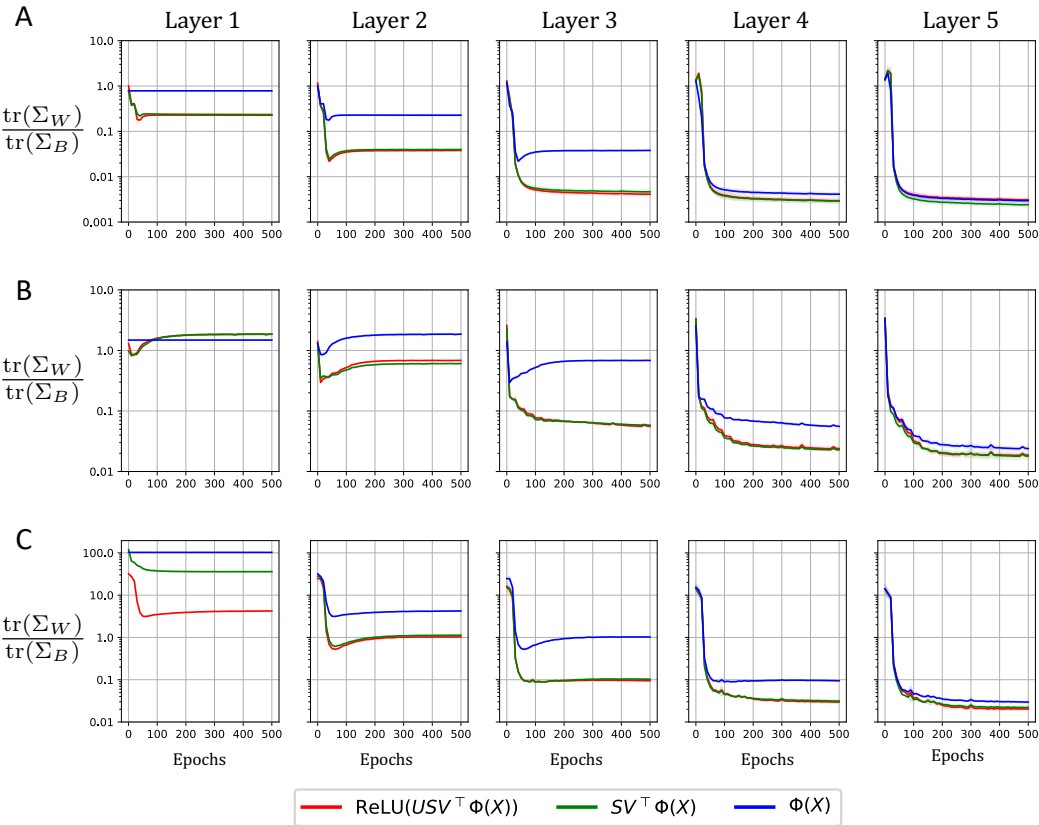

Figure 7: Feature variability collapse (NC1) from different singular value decomposition components on an MLP. The first row (A) is MNIST, second (B) is CIFAR-10, third (C) is SVHN. As in the main text, we plot the NC1 metrics, for the original feature vectors $\Phi(X)$ (blue), the data after applying the right singular structure (green), and the data after the full layer application (red).

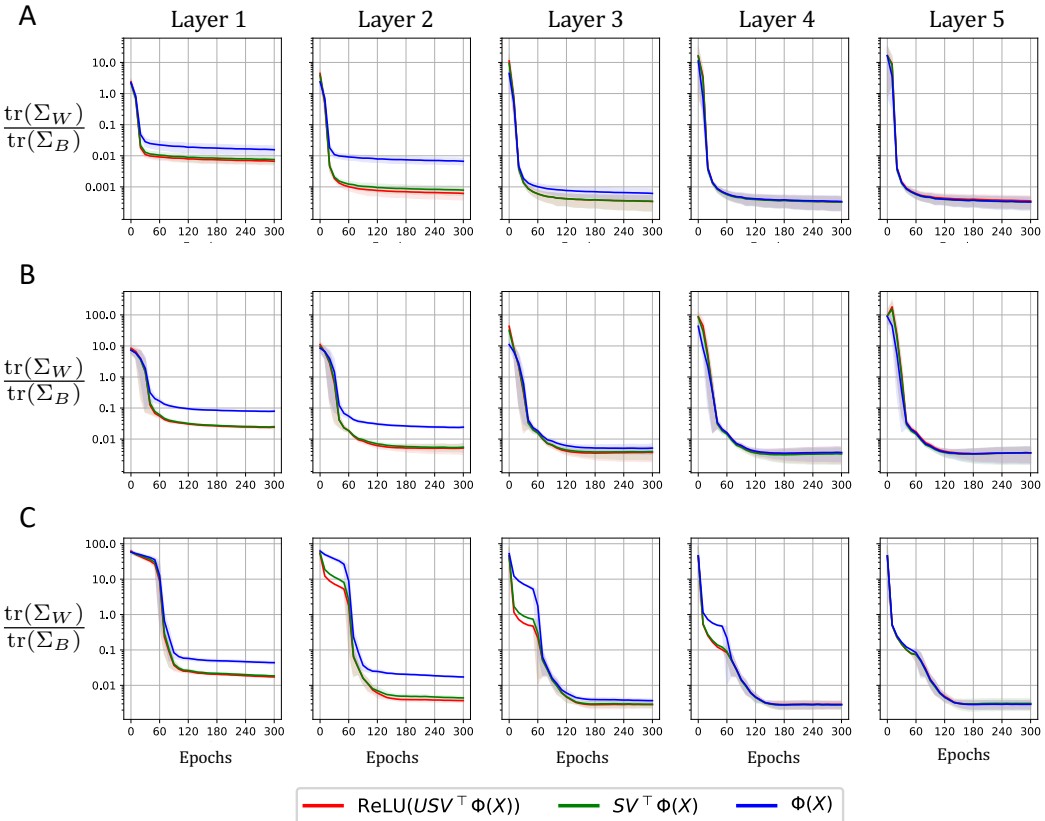

Figure 8: Feature variability collapse (NC1) from different singular value decomposition components on ResNet. The first row (A) is MNIST, second (B) is CIFAR-10, third (C) is SVHN. As in the main text, we plot the NC1 metrics, for the original feature vectors $\Phi(X)$ (blue), the data after applying the right singular structure (green), and the data after the full layer application (red).

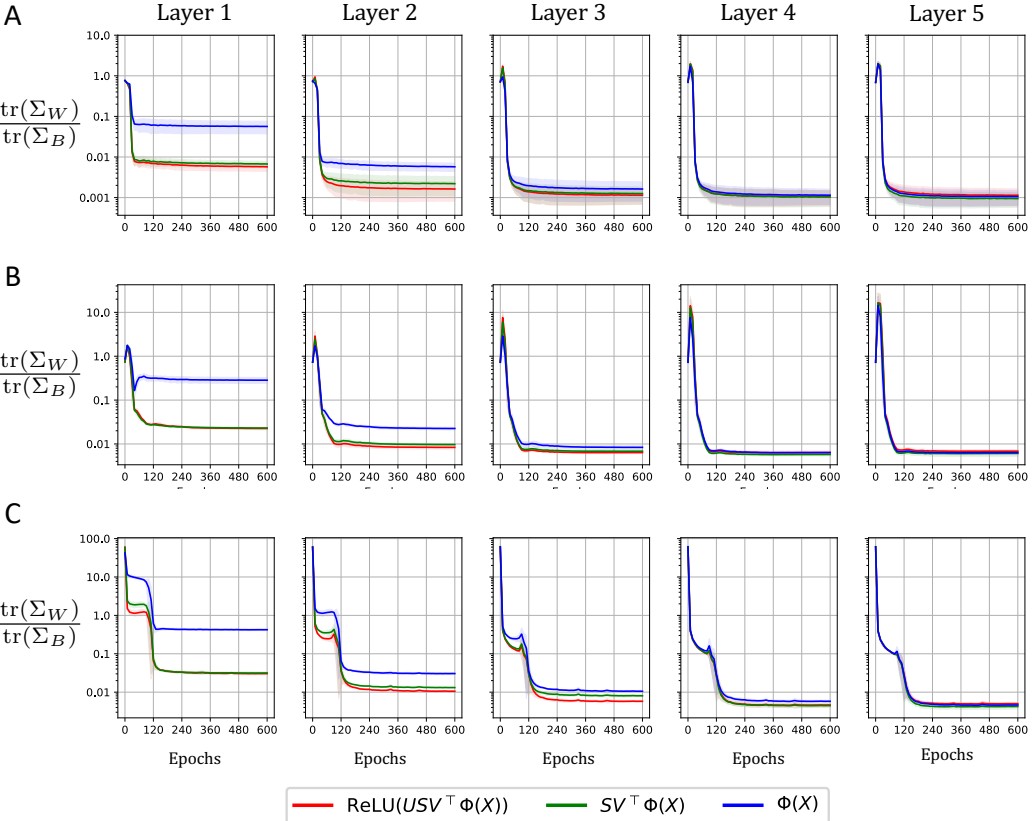

Figure 9: Feature variability collapse (NC1) from different singular value decomposition components on VGG. The first row (A) is MNIST, second (B) is CIFAR-10, third (C) is SVHN. As in the main text, we plot the NC1 metrics, for the original feature vectors $\Phi(X)$ (blue), the data after applying the right singular structure (green), and the data after the full layer application (red).

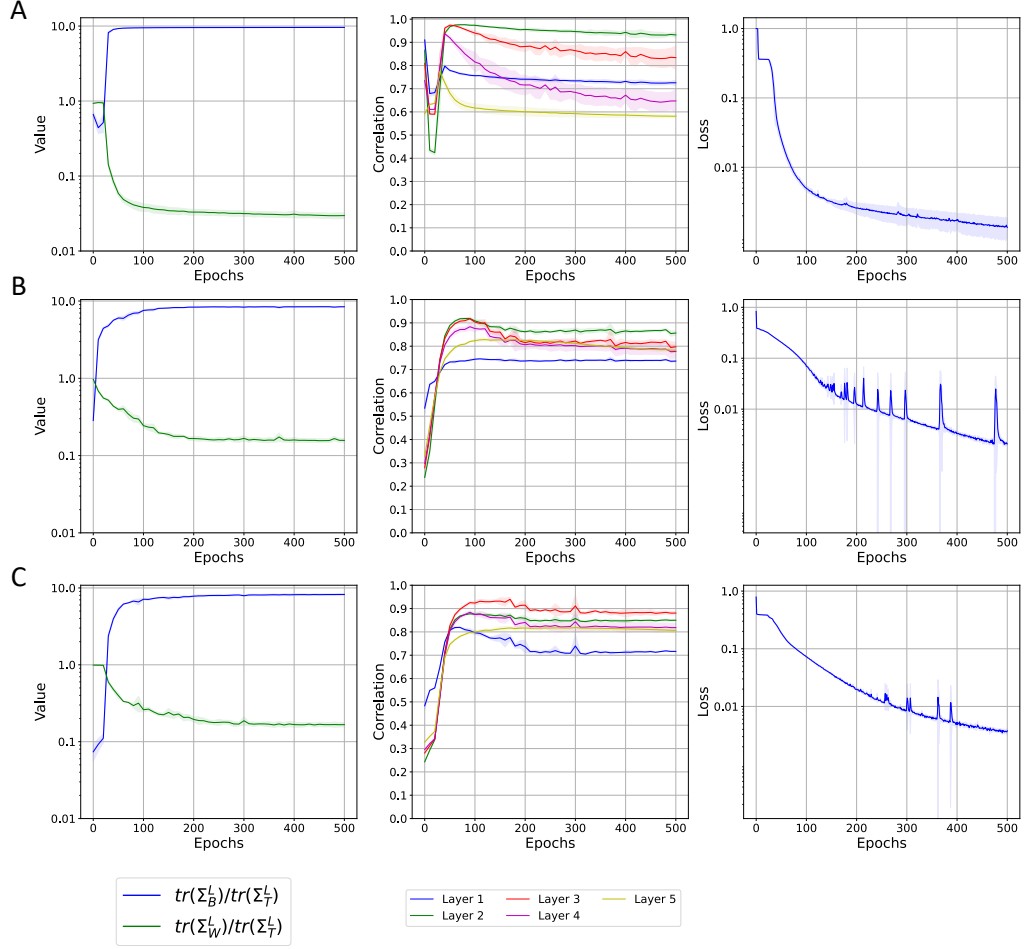

Figure 10: Train loss and NFA correlations for MLPs. The first row (A) is MNIST, second (B) is CIFAR-10, third (C) is SVHN. In the first column, we plot the evolution of $\operatorname{tr}\Sigma_W/\operatorname{tr}\Sigma_T$ and $\operatorname{tr}\Sigma_B/\operatorname{tr}\Sigma_T$, where $\Sigma_T \triangleq \Sigma_W + \Sigma_B$. In the second column we plot the development of the NFA, where correlation is measured between $W_l^\top W_l$ and the square root of the AGOP with respect to the inputs at layer $l$, $X_l$. The third column is the train loss of the neural network.

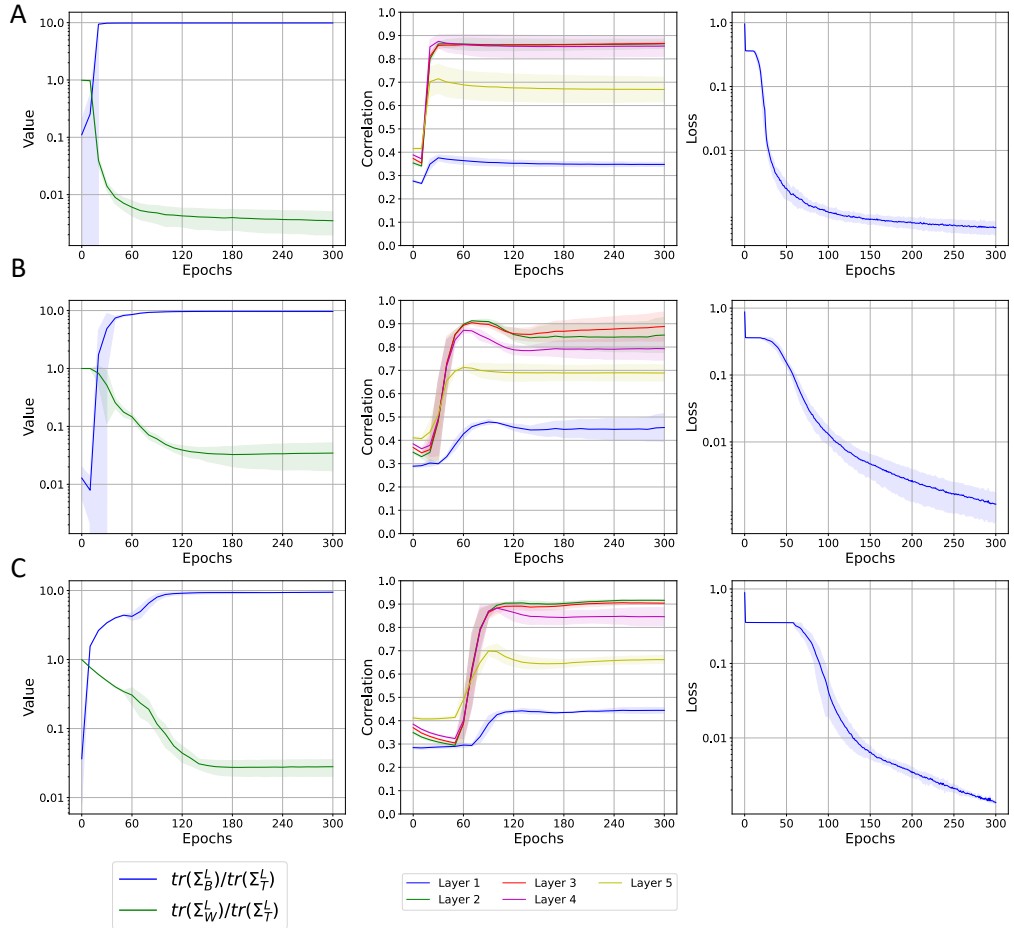

Figure 11: Train loss and NFA correlations for ResNet. The first row (A) is MNIST, second (B) is CIFAR-10, third (C) is SVHN. In the first column, we plot the evolution of $\operatorname{tr}\Sigma_W/\operatorname{tr}\Sigma_T$ and $\operatorname{tr}\Sigma_B/\operatorname{tr}\Sigma_T$, where $\Sigma_T \triangleq \Sigma_W + \Sigma_B$. In the second column we plot the development of the NFA, where correlation is measured between $W_l^\top W_l$ and the square root of the AGOP with respect to the inputs at layer $l$, $X_l$. The third column is the train loss of the neural network.

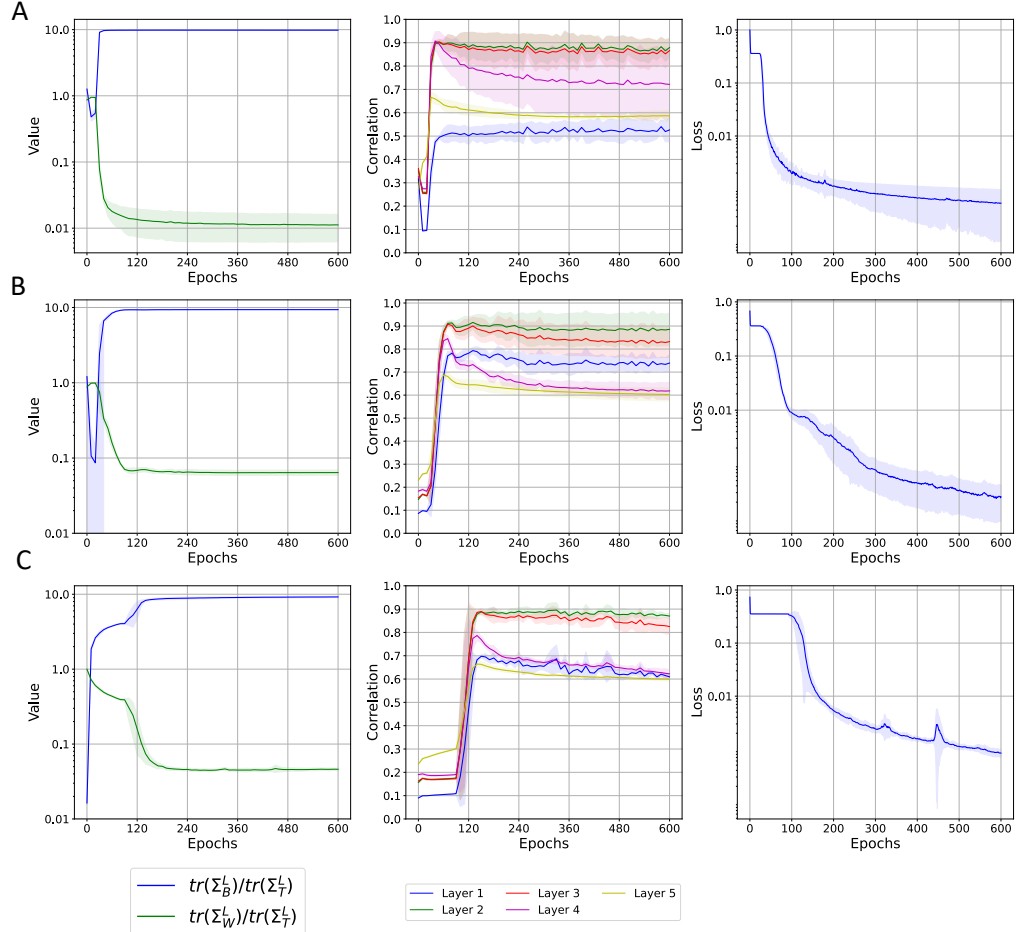

Figure 12: Train loss and NFA correlations for VGG. The first row (A) is MNIST, second (B) is CIFAR-10, third (C) is SVHN. In the first column, we plot the evolution of $\operatorname{tr}\Sigma_W/\operatorname{tr}\Sigma_T$ and $\operatorname{tr}\Sigma_B/\operatorname{tr}\Sigma_T$, where $\Sigma_T \triangleq \Sigma_W + \Sigma_B$. In the second column we plot the development of the NFA, where correlation is measured between $W_l^\top W_l$ and the square root of the AGOP with respect to the inputs at layer $l$, $X_l$. The third column is the train loss of the neural network.

