# OpenReview forum: "Average gradient outer product as a mechanism for deep neural collapse"
_NeurIPS.cc/2024/Conference — NeurIPS 2024 poster_

### Official Review · Reviewer_Ljpy · 2024-07-08

**Soundness:** 3
**Presentation:** 3
**Contribution:** 2
**Rating:** 5
**Confidence:** 3

**Summary:**

Given the complexity of the process of neural network training, any understanding of robust phenomena that can be identified in the training process has potential value that can guide the design of models and algorithms. Neural Collapse (and its deep counterpart) is one such phenomenon that has been identified and reproduced across multiple model classes and datasets. This work shows that Neural Collapse also occurs for a recursive kernel-based model known as Deep RMF, when trained using an algorithm that is based on projection onto a matrix constructed from an outer products of gradients computed locally at each layer.
Additionally, the authors present experimental results that document neural collapse in these models when trained on standard datasets. They also show that in standard neural networks, the projection of features onto the gradient outer product leads to neural collapse, rather than the effect of the nonlinearity.

**Strengths:**

The paper is clearly written, and presents both theoretical results and some empirical results that complement them, since they apply to datasets that violate the assumptions under which the results hold. They prove that deep Neural Collapse can indeed occur in models beyond standard neural networks trained with gradient descent.
The experimental results (specifically in Appendix D) demonstrate that the projection onto the gradient outer product matrix (AGOP) leads to neural collapse in standard models, motivating the further study of this object.

**Weaknesses:**

Given that the main results apply both to a non-standard kernel method and a non-standard training algorithm, it is unclear what the implications of the results are for more well-known models and algorithms. If the authors believe that these results have implications of this form, they should be presented more clearly. Algorithms that are not based on backpropagation are interesting both as possible means of explaining learning in biological systems where backpropagation is unrealistic, and in distributed settings where backpropagation may incur a prohibitive communication overhead. However, the motivation of the algorithm used appears to be that it is a simple model that demonstrates certain phenomena that arise in the training of deep networks.

The authors assume that the gram matrix of the data is full-rank. This requires assuming that the number of datapoints is smaller than or equal to the input dimension (which subsumes assumption 4.1). Standard datasets violate this assumption.

**Questions:**

Are the models in appendix D trained with SGD? If so, they indicate the importance of the AGOP projection in causing neural collapse in standard models and I believe this result should be highlighted. That being said, this result may be of independent interest regardless of the analysis of Deep RMF.

**Limitations:**

Limitations and societal impacts have been addressed

---

> ### Author Rebuttal · Authors · 2024-08-01
>
> Thank you for your review. We address all your concerns below:
>
> **Given that the main results apply both to a non-standard kernel method and a non-standard training algorithm [...]**
>
> Our first motivation for studying DNC with Deep RFM is that, unlike the standard analytical approaches for neural collapse, the Deep RFM procedure is highly dependent on both the input samples and their labels. Specifically, the collapse process of Deep RFM depends on mapping with the AGOP of a predictor trained to fit these input/label pairs. Moreover, as we demonstrate in Theorem 4.3, the rate of this collapse throughout the layers is highly dependent on the specific structure of the data through its relationship to the parameters $\lambda_\Phi$ and $\lambda_{\mathrm{lin}}$. We also provide significant evidence that this AGOP mechanism is present in deep neural networks in Section 5 of our original manuscript, indicating that our results apply to more practical architectures.
>
> Secondly, the ability of Deep RFM to recover interesting behaviors of deep learning has broad implications for our understanding of neural networks. In fact, the same Deep RFM model we study has recovered a number of other interesting deep learning phenomena including, e.g., grokking modular arithmetic [1], convolutional feature learning [2], and even low-rank matrix completion [3]. As you mention, Deep RFM is a simpler model than neural networks, in the sense that we understand its individual components: (1) a kernel regression step, (2) mapping with the AGOP, and (3) random feature projection. Therefore, our work additionally justifies Deep RFM as a simplified proxy to understand neural networks.
>
> [1] Mallinar, Beaglehole, Zhu, Radhakrishnan, Pandit, Belkin, “Emergence in non-neural models: grokking modular arithmetic via average gradient outer product.” arXiv pre-print, 2024.
>
> [2] Radhakrishnan, Beaglehole, Pandit, Belkin, "Mechanism for feature learning in neural networks and backpropagation-free machine learning models." Science, 2024.
>
> [3] Radhakrishnan, Belkin, Drusvyatskiy, "Linear Recursive Feature Machines provably recover low-rank matrices." arXiv pre-print, 2024.
>
> **The authors assume that the gram matrix of the data is full-rank. This requires assuming that the number of datapoints is smaller than or equal to the input dimension (which subsumes assumption 4.1). Standard datasets violate this assumption.**
>
> The strong assumption that the gram matrix of the data is full-rank is needed only if one requires neural collapse in every layer of the network, starting from the very first one. In contrast, if we consider collapse starting at any given later layer of Deep RFM, then we only need that the smallest eigenvalue of the corresponding feature map is bounded away from 0. This in turn requires that the number of features at that layer is greater than the number of data points, a condition which is routinely satisfied by the overparameterized neural networks used in practice.
>
> **Are the models trained with SGD?**
>
> The neural network models are trained with standard SGD. We agree that this result is of independent interest, and we will emphasize in the main section that our training procedure is standard. Let us also clarify that Deep RFM is not trained by any gradient-based method, and is instead optimized through an iterated three-stage process: (1) solve kernel ridgeless regression, (2) compute the AGOP of the predictor from step 1, and (3) apply a high-dimensional feature map.

---

> > ### Comment · Reviewer_Ljpy · 2024-08-08
> >
> > I would like to thank the authors for their detailed response. After consideration of their comments and those of the other reviewers I have decided to increase my score.

---

### Official Review · Reviewer_uXDR · 2024-07-10

**Soundness:** 3
**Presentation:** 1
**Contribution:** 3
**Rating:** 6
**Confidence:** 3

**Summary:**

This paper studies deep neural collapse (DNC) in deep neural networks (DNN) through the prism of the neural feature ansatz (NFA) and deep recursive feature machines (RFM). It is comprised of several results:
- empirical evidence that DNC occurs in deep RFMs,
- a theoretical analysis of DNC in a high-dimensional RFM setting,
- a theoretical analysis of DNC in a kernel learning setting,
- empirical evidence that the mechanisms which lead to DNC in RFMs and traditional DNNs are the same.

**Strengths:**

This paper shows that deep neural collapse occurs in a similar way in deep networks and deep recursive feature machines. It thus provides a simplified setting in which to investigate deep neural collapse, which is an important research direction to further our understanding of deep learning. Specifically, it shows that neural collapse can be obtained just by iterating linear regression problems, without backpropagating through a deep network.

**Weaknesses:**

My main issue with the paper is its writing, which makes it quite difficult to read.
- The notations could be improved in several places throughout the paper (see minor points below).
- I could not follow most of section 4.2, despite being rather familiar with kernel methods and their behavior in high dimensions.
On a high level, I don't understand how a linear kernel could be the best setting for neural collapse. The text contradicts itself, as it simultaneously state that "if [$\lambda_k = 0$] [...], collapse will occur in just one layer] , but also that "this theory offers an explanation for why non-linear activation is needed". A linear layer can collapse within-class variability but also typically collapses class means together, and thus cannot induce neural collapse (see paragraph below).
On a technical level, $k_\Phi$ and $\lambda_\Phi$ are referred to before being defined, and I do not understand the roles played by $k$/$\lambda_k$ vs $k_\Phi$/$\lambda_\Phi$. Assumption 4.2 is also referred to before being stated.
- Section 4.3 is also slightly difficult to read.
I took me several tries to guess that $k_M(x,x') = \tilde k_M(x,x')\mathrm{Id}$, which should appear in the paper. The terms "input-level" and "output-dimension level" kernels should be introduced for non-specialists in multi-task kernel learning.
I also do not understand the point of introducing $M$ if it is dropped afterwards. Theorem 4.4 could simply be stated as "the optimal feature map for ridge regression is the one which already predicts the label: $\Phi(x) = y$". This result is not very surprising, and is not very integrated in the paper. I suppose that it is some kernel-level equivalent of the unstructured feature model, and suggests that weight decay might be instrumental in bringing about neural collapse? The normalization of $k$ should be restated in the definition of Problem 3 (otherwise the optimal loss is obtained when $k \to 0$).
- The message of section 5 could be presented more clearly. What I understood was that it argues that RFMs and DNNs achieve neural collapse through the same means. I suggest making this point before introducing RFMs (in particular, stating the NFA correlations). I also did not understand why this mechanism is referred to as "denoising".

My second issue is that I was not convinced by the claim that it is the right singular vectors and singular values which lead to neural collapse. By the same logic as lines 309-315, the right singular vectors do not change the DNC1 metric (with a "full" SVD where $U$ and $V$ are invertible). Similarly, if I were to divide operations in the network as $V^T\sigma$ and $US$ as opposed to $\sigma U$ and $SV^T$, I should see that it is now $US$ which is responsible for neural collapse (again with a full SVD). This conclusion also depends on the chosen metric for evaluating collapse. Why do the authors consider the ratios of traces of between- and within-class covariances, rather than the trace of their ratio (the Fisher linear discriminant)? It seems that it would reverse one of the conclusions of the analysis, since the trace of the Fisher discriminant ratio $\mathrm{tr}(\Sigma_W^{-1} \Sigma_B)$ is invariant to invertible linear transformations, and decreases under non-invertible linear transformations, so can only be improved through the non-linearity. If the conclusion of which part of the network is responsible for DNC depends strongly on the chosen metric, can we really ascribe meaning to the question? It seems to me that it is really the sequence of weights and non-linearity which _together_ induce DNC, and trying to separate their effects is not really possible.

Finally, Proposition A.1 was first published by Cho and Saul in _Kernel Methods for Deep Learning_, NIPS 2009. Besides, the expression of the kernel in eq. (5) can be simplified with algebra and trigonometry (compare with their eq. (6)).

Minor notes and suggestions:
- I suggest using a so-called "diverging" colormap (such as "bwr") in Figure 1 to clearly separate positive from negative correlations, and use the same range for both datasets.
- I suggest replacing "Gram matrix" with "(uncentered) covariance" to refer to $W^TW$, as weight matrices $W$ are generally decomposed in rows which correspond to individual neurons.
- The notation $||X||$ to refer to the vector in $\mathbb R^N$ of column norms of a matrix $X \in \mathbb R^{d\times N}$ is never introduced (and clashes with the usual convention that this is a matrix norm).
- Why is the last layer denoted $W_{L+1}$ instead of $m_{L+1}$?
- The choice of layer-indexing is confusing and seems inconsistent throughout the paper. Contrarily to what is stated in section 3.1, isn't $X_l$ the features after $l-1$ network layers? I suggest to denote the input as $X_0$ instead of $X_1$ to simplify the notations. Also, it seems that $M_l^{1/2} X_l$ should be referred to as $\tilde X_{l+1}$ rather than $\tilde X_l$ given the chosen conventions.
- Typo: missing a norm in the definition of $\bar H_l$ line 128.
-In section 4.2, I suggest defining activations before the kernels, e.g., $\tilde X_{l+1} = \kappa^{-1/2} M_l^{1/2} X_l$ and $X_{l+1} = \Phi_{\rm lin}(\tilde X_{l+1})$. I also suggest choosing a different notation for $k_{\rm lin}$ and $\Phi_{\rm lin}$ which are confusing as they imply linearity, and to avoid the awkward "non-linear feature map $\Phi_{\rm lin}$".
- Typo line 260: the output space of $k_M$ should be $\mathcal R^{C\times C}$.
- I suppose that $\lambda = \mu$ in section 4.3.
- In the caption of Figure 2, I suppose that "fully-connected" should be removed in the case of ResNet.

**Questions:**

- Are the feature maps $\Phi_l$ and kernels $k_l$ unrestricted in Algorithm 1, or do they have to match in the sense that $k_l(x,x') = \langle \Phi_l(x), \Phi_l(x')\rangle$?
- What is the motivation behind considering _normalized_ features _before_ the non-linearity in section 4.1? Could the authors clarify the role of these non-standard conventions?
- Why are the setting different between sections 4.1 and 4.2? (relative to normalizations). Is neural collapse still empirically observed with the choices of section 4.2? It raises the suspicion that it could not the case, in which case Theorem 4.3 would not really explain the results of section 4.1 (e.g., because the asymptotic regime is not reached in practice).

**Limitations:**

See weaknesses.

In its current state, I think that the paper is slightly below the acceptance bar and would require minor, if not major, changes before it can be fully appreciated by the NeurIPS community. I would be happy to raise my score if the authors address the points raised above.

---

> ### Author Rebuttal · Authors · 2024-08-01
>
> Thank you for your feedback. We note that we will significantly improve the presentation of our paper, and specifically Sections 4.2 and 4.3. Please see the global response for a summary of our changes. We now proceed to address individual comments.
>
> **I could not follow most of section 4.2 [...]**
> We will clarify this. The two kernels $k_{\mathrm{lin}}$ and $k_{\Phi}$ have opposite effects. In particular, the closer $k_{\mathrm{lin}}$ is to a linear kernel, the more accelerated collapse will be for Deep RFM. In fact, a linear kernel induces NC in just one AGOP application. To see this, note that ridgeless regression with a linear kernel is exactly least-squares regression on one-hot encoded labels. In the high-dimensional setting we consider, we find a linear solution $f(x) = \beta^T x$ that interpolates the labels, and the AGOP of $f$ is $\beta \beta^T$. Since we interpolate the data, $\beta^T x = y$ for all $(x,y)$ input/label pairs and, hence, the data collapses to $\beta\beta^T x = \beta y$.
>
> In contrast, having $k_{\Phi}$ far from a linear kernel accelerates collapse, and the non-linear activation function is needed so that $k_{\Phi}$ is significantly non-linear. In fact, if $k_{\Phi}$ were exactly linear and the original data were not linearly separable, then collapse cannot occur at any depth of Deep RFM. In that case, the predictor would need to contain a non-linear component in order to fit the labels exactly, deviating from the ideal case described above.
>
> **Section 4.3 is also slightly difficult to read [...]**
> Thank you for your suggestions. We address your points one-by-one:
>
> - We introduce $M$ because it is the key part of the parametrized kernel ridge regression which can be understood as the objective function of the RFM iteration. We emphasize that when the data (or the features) are full rank, dropping $M$ does come without loss of generality. Due to the character limit, we kindly refer you to our response to reviewer AtBe for details.
> - We appreciate your interpretation of this model as an unconstrained kernel model. We agree with this interpretation and will call it in this way in the revision.
> - Theorem 4.4 is well integrated in our paper as the kernel ridge regression loss can be understood as the quantity the RFM learning algorithm is minimizing. Thus, the result describes DNC as an implicit bias of the RFM iteration.
> - For additional structural changes to Section 4.3, see our global response.
>
> **The message of section 5 [...]**
> “Linear denoising” highlights the nature of the mechanism: by only applying linear transformations, we decrease the within-class variability of the data and improve DNC1 by only discarding class-irrelevant information (i.e., noise) via the non-trivial null-space of the weight matrix. See our global response for additional changes.
>
> **Why splitting into $\sigma U$ and $SV^T$?**
> In the compact SVD (which is an equivalent re-writing of the full SVD), it is the matrix $V^T$ which has non-trivial null-space, while the matrix $U$ always has a trivial null-space. Note that the non-triviality of the null-space allows for the DNC1 metric to decrease. Hence, this is the correct split of the layer.
>
> More generally, the main message of this section is not which part of the SVD decreases the DNC1 metric, but rather that the linear transformation of $W$ has a stronger effect on NC1 than the non-linearity $\sigma$. In fact, the most natural grouping of the layer - into the non-linearity alone and the entire weight matrix alone - gives equivalent NC1 values to our grouping, as the two standard metrics for NC1, $\mathrm{tr}(\Sigma_W \Sigma_B^\dagger)$ and $\mathrm{tr}(\Sigma_W)/\mathrm{tr}(\Sigma_B)$, are invariant to the rotation by the left singular vectors of $W$.
>
> **This conclusion also depends on the chosen metric...**
> This is an interesting consideration, however we kindly disagree that Fisher linear discriminant cannot decrease due to a linear mapping. Consider the example in which the feature vectors of $X$ are not collapsed, but the differences between points of the same class lie in the null-space of $W.$ Then, $WX$ is perfectly collapsed and any NC1 metric would be identically zero.
>
> While the two metrics are similar, we choose our metric because it directly quantifies how close the feature vectors are to their class means - a common and intuitive interpretation of NC1. Our metric is also frequent in the literature, see e.g. Rangamani et al., 2023 and Tirer et al., 2023.
>
> **Are the feature maps $\Phi_l$ and kernels $k_l$ unrestricted?**
> The feature maps and kernels do not have to match. We will clarify this when we introduce the two objects.
>
> **What about the normalized features in 4.1?**
> We normalize the features as neither the AGOP projection nor the random feature map have a native ability to rescale the data. Therefore, while we still see collapse in terms of the angles between data points, the dynamics of the data norms are more difficult to control in this setting.
>
> **Why are the setting different between sections 4.1 and 4.2? [...]**
> Neural collapse would still be observed without normalizing the data, in the sense that all points of the same class will be parallel and all points of different classes will be perpendicular (or at angle $-\frac{1}{K-1}$). However, we are not guaranteed convergence of the data Gram matrix to $yy^T$ in general without the re-normalization, as the diagonal entries of $X^T X$ will not be identically $1$. In our analysis, it is sufficient to scale the Gram matrix at each layer with $\kappa^{-1}$. The main advantage of this scaling is that it is analytically much simpler than explicitly projecting the data to be norm $1$.
>
> **Minor notes.**
> We will address all these points in the revision. As an example, we will rename $k_{\mathrm{lin}}$ and $\Phi_{\mathrm{lin}}$ to $\widehat{k}$, indicating the predictor kernel, and $\Phi_{\mathrm{map}}$, indicating the feature map applied to the data.

---

> > ### Comment · Reviewer_uXDR · 2024-08-07
> >
> > I thank the authors for their detailed response. They have addressed my main concern regarding the writing of the paper. As a result, I have increased my score.
> >
> > If I understand correctly, in this paper neural collapse is only considered on training data? That is, generalization is not required and a perfectly overfitting network would achieve neural collapse? I think this should be explicitly pointed out in the text. Indeed, it makes it very easy to achieve neural collapse, with a number of random features that is larger than the number of training samples, followed by a linear layer. Doesn't the statement "deep RFMs can achieve neural collapse" then reduce to "deep RFMs can overfit their training set"? A mystery of neural collapse that is not studied here is that the within-class variability remains negligible _on the test set_.
> >
> > Thank you for pointing out that the Fisher linear discriminant can decrease under noninvertible linear transformation, I was mistaken. I however maintain that stating something like "the linear layers are mostly responsible for neural collapse" is misleading, as removing all non-linearities from the network would prevent neural collapse from arising since the data is not non-linearly separable. So the non-linearities must have a role, as they "enable" the collapse of within-class variability by the next linear layer. Although this is not reflected in the metrics evaluated at intermediate points, non-linearities thus play an important role. To iterate on the question of the metric, if I replace NC1 with "the value of NC1 after applying a linear operator that minimizes NC1" (i.e. after a linear "probe" classifier), then only non-linearities can decrease this new metric.

---

> ### Author Response · Authors · 2024-08-08
>
> We thank the reviewer for their reply.
>
> Yes, we only study neural collapse on the training set. Note that there are mixed empirical results about whether or not neural collapse transfers to the test set [1, 2, 3]. Moreover, to the best of our knowledge no work has studied the emergence of neural collapse at test time theoretically. Perfect collapse at test time would mean perfect generalization, therefore only weaker forms of test-collapse are available [3].
>
> We emphasize that neural collapse is a particularly structured form of interpolation that is not implied by low train loss, or overfitting, alone. In fact, there are many ways either neural networks or Deep RFM can interpolate the training data. Therefore, it is remarkable that the particular interpolating solutions found by these models exhibit DNC. Our paper, as well as all the other theoretical papers on collapse, study why among all interpolations for a given model class and dataset, the training procedure selects a solution exhibiting DNC.
>
> You are correct, the non-linearity is essential to achieve the collapse for linearly non-separable data. The ReLU plays an important role, as your proposed metric shows. We will make it clear in our revision that what we mean by saying that the linear layers are responsible for reducing within-class variability is not that they can achieve collapse on their own, but that in the solved model, it is the linear layers that directly decrease the NC1 metric. Moreover, the ReLU plays a critical role in enabling collapse, especially when the data is not linearly separable.
>
> [1] Xu and Liu. “Quantifying the variability collapse of neural networks.” International Conference on Machine Learning, 2023.
>
> [2] Kothapalli. “Neural collapse: A review on modelling principles and generalization.” arXiv pre-print, 2022.
>
> [3] Hui, Belkin, Nakkiran. “Limitations of neural collapse for understanding generalization in deep learning.” arXiv pre-print, 2022.

---

> > ### Comment · Reviewer_uXDR · 2024-08-08
> >
> > Thank you very much for these precisions and the references. I had wrongly assumed that neural collapse was known to occur on the test set, but I see the picture is much more nuanced.

---

### Official Review · Reviewer_1F1v · 2024-07-12

**Soundness:** 2
**Presentation:** 2
**Contribution:** 2
**Rating:** 6
**Confidence:** 2

**Summary:**

The submission introduces a mechanism for Deep Neural Collapse (DNC) using the average gradient outer product (AGOP). The authors also propose the Deep Recursive Feature Machines (Deep RFM) model, which employs AGOP in its architecture to empirically and theoretically demonstrate DNC. The main contribution is that AGOP-based explanation is a data-based approach while prior work focused on data-agnostic explanations.

**Strengths:**

* Using a data-based approach based on AGOP to explain DNC is novel to the best of my knowledge
* The paper offers both theoretical analysis and empirical evidence supporting the role of AGOP in inducing DNC
* The experiments are performed on different architectures and datasets

**Weaknesses:**

*  I found the paper challenging to read
* I am unsure about the practical implications of this work

**Questions:**

* Can the authors clarify if other metrics, such as the Neural Tangent Kernel (not the limit), would effectively predict this behavior? Or AGOP is unique in this aspect?
* What are the practical implications of this work?

**Limitations:**

See weaknesses.

---

> ### Author Rebuttal · Authors · 2024-08-01
>
> Thank you for your review. We address your concerns below.
>
> **I found the paper challenging to read.**
>
> We will make a number of changes to our presentation. Please see our global response to all reviewers for a summarized description of these changes.
>
> **Can the authors clarify if other metrics, such as the Neural Tangent Kernel (not the limit), would effectively predict this behavior? Or AGOP is unique in this aspect?**
>
> Prior work has shown that neural collapse can occur if the empirical Neural Tangent Kernel has a particular block structure dependent on the inputs and labels [1]. In Deep RFM, we consider a different setting, where we fix a kernel function to be independent of the data, i.e. the NTK in the infinite-width limit, and then understand how a linear transformation prior to the kernel evaluation can induce DNC.
>
> [1] Seleznova, Weitzner, Giryes, Kutyniok, Chou. “Neural (Tangent Kernel) Collapse.” NeurIPS, 2023.
>
> **What are the practical implications of this work?**
>
> While the scope of this work is theoretical, there are a number of works demonstrating the practical applications of (deep) neural collapse and Recursive Feature Machines. More specifically, neural collapse has been used as a tool in several important contexts, including generalization bounds, transfer learning, OOD detection, network compression and robustness, see lines 85-90 and the references therein. Furthermore, RFM itself has been shown to be a practical tool for tabular data (Radhakrishnan et al., Science 2024) and scientific applications [1,2].
>
> Our work also has broad implications for our understanding of deep learning. In particular, this work is part of a large body of research demonstrating that Deep RFM is a useful proxy to explain interesting phenomena in deep networks. In addition to neural collapse considered in this work, RFM has been shown to exhibit grokking in modular arithmetic [3], recover convolutional edge detection, and learn the same features as fully-connected networks on vision tasks (Radhakrishnan et al., Science 2024). These findings point toward Deep RFM and AGOP as powerful tools to improve our understanding of neural networks, which would likely lead to significant practical implications.
>
> [1] Aristoff, Johnson, Simpson, Webber. “The fast committor machine: Interpretable prediction with kernels.” arXiv pre-print, 2024
>
> [2] Cai, Radhakrishnan, Uhler, “Synthetic Lethality Screening with Recursive Feature Machines.” arXiv pre-print, 2023.
>
> [3] Mallinar, Beaglehole, Zhu, Radhakrishnan, Pandit, Belkin, “Emergence in non-neural models: grokking modular arithmetic via average gradient outer product.” arXiv pre-print, 2024.

---

> > ### Comment · Reviewer_1F1v · 2024-08-08
> >
> > I thank the authors for their response. They have addressed most of my concerns. Therefore, I have increased my score. I look forward to the updated version of the manuscript.

---

### Official Review · Reviewer_AtBe · 2024-07-13

**Soundness:** 4
**Presentation:** 3
**Contribution:** 4
**Rating:** 6
**Confidence:** 4

**Summary:**

The authors study two effects associated with neural collapse: the within class variability going to zero and the orthogonality/tight-frame of the class means. They study the deep recursive feature machine model, and show that neural collapse forms in that setting as well, due to the projection of the data onto the average-gradient outer product (AGOP). They show both empirical and theoretical results on this phenomenon, leveraging high-dimensional gaussian equivalence of nonlinear random feature models. Further, they show that the right singular vectors of the weight matrices are responsible for most the within-class variability collapse, projecting onto the subspace spanned by the gradients.

**Strengths:**

The writing of the paper is for the most part quite readable. The literature review is thorough and puts the results of this paper in a good context. The empirics are extensive and compelling. Moreover, the theoretical ideas leveraging the equivalence of nonlinear random features to a linear kernel with additional identity term make for a compelling argument about the mechanism for neural collapse in RFMs. Given the good mix of theory and experiment, I recommend this paper for acceptance.

**Weaknesses:**

Section 4 is doing many things at the same time. It may be better to split it into an empirical evidence section, and then do a section on the theoretical results. In particular, it would be good to give an idea of where the theoretical ideas are going at the start of 4.2 before setting out to prove the deep neural collapse results. This would substantially improve the readability of this section.

This goes double for section 4.3. The opening paragraph of that section is unreadable:

*"Next, we show that the formation of the neural collapse is not only implicitly given by the specific
optimization procedure of the Deep RFM, but is also implicitly regularized for in the parametrized
kernel ridge regression, a model class that includes RFM (i.e., a single layer of Deep RFM)"*

I don't really understand what this is saying, or even what you're trying to accomplish in the entire subsection. I tried many times to read it. The whole subsection should be rewritten. There are many sentences there that make no sense to me. Here is another one:

*"Since we do not explicitly regularize M, we will drop the dependence on it, treat k as
 a free optimization variable and compute the optimal value of the following relaxed problem:"*

This is certainly not something you can do generally. For example if I had a matrix parameterized as $A = M_1 M_2$ and optimized just the $M_i$ with no explicit regularization, there are many cases where this isn't the same as optimizing $A$. Maybe you mean to say something else but once again I can't understand what you're trying to say. The prior subsection was compelling enough that I am discounting this rather poor form of writing. Please rewrite this section.

More generally, there are many sentences throughout the paper that are not well-worded and seem to run on. Improving the writing would benefit the quality, precision, and reach of this otherwise strong paper. If in your rebuttal you can provide examples of improved presentation, I may raise my score higher.

**Questions:**

One can show that for a $\ell_2$ regularized deep network that the weights pick up rank one spikes proportional to $W^{\ell}_{ij} \propto  \frac{\partial f}{\partial x^\ell_i} \phi(x^{\ell}_j)$ where $\phi$ is a nonlinearity and $x^\ell$ is the preactivation.. This usually means that the *left* singular values of the weight matrices should pick up terms aligned with $\nabla f$. See for example the update equation for $W$ in 2.2.4 of LeCun:

http://yann.lecun.com/exdb/publis/pdf/lecun-88.pdf

Is there any easy way to square this with the results on RFMs, that the *right* singular values align with $\nabla f$ terms?

It would be good to be explicit and put a subscript below the $\nabla$s on the $\nabla f_\ell(x^\ell_{c i})$ in algorithm 1 to be clear what you're differentiating with respect to.

I don't understand why 4.3 is called non-asymptotic analysis. I don't think you're proving non-asymptotic bounds compared to 4.2. If anything 4.3 seems completely unrelated. Can you please give it a title where someone can understand what you're trying to do? Once again the rest of the paper is quite readable but this subsection is a mess.

**Limitations:**

This work elucidates an important phenomenon in deep learning theory. Developing a principled understanding of feature learning is likely to have implications for the interpretability and reliability of AI systems.

---

> ### Author Rebuttal · Authors · 2024-08-01
>
> Thank you very much for your detailed review. In our response here, we will pay significant attention to improving the writing and organization of our work, especially Section 4. Please also read our global response where we discuss the changes in presentation in detail to all reviewers. We proceed by addressing each weakness and question individually.
>
> **Section 4 is doing [...]**
>
> Thank you for this suggestion, please see our global response for the changes we will make in our revision.
>
> **The opening paragraph of Section 4.3 is unreadable.**
>
> We will present our result differently. As an example of our improved presentation, we would rewrite the opening paragraph of this section as follows:
>
> “We have shown so far that mapping with the AGOP induces DNC in Deep RFM empirically and theoretically in an asymptotic setting. In this section, we demonstrate that DNC emerges in parameterized kernel ridge regression: in fact, DNC arises as a natural consequence of minimizing the norm of the predictor jointly over the choice of kernel function and the regression coefficients. This result proves that DNC is an implicit bias of kernel learning. We connect this result to Deep RFM by providing intuition that the kernel learned through RFM effectively minimizes the parametrized kernel ridge regression objective and, as a consequence, RFM learns a kernel matrix that is biased towards the collapsed gram matrix $yy^T$.”
>
> **Dropping the dependence on $M$ is not always equivalent to the original problem.**
>
> You are right that optimizing over all $k$ is not always equivalent to optimizing over all $M$. Thus, in general, this provides only a relaxation. However, if the gram matrix of the data is invertible (this is a realistic assumption if we consider layers $\ell>1$ of Deep RFM, where we are allowed to pick the feature dimension to be large enough so that this is satisfied), then the two optimizations have the same solution (where the min is taken to be an infimum) and, thus, the relaxation is without loss of generality.
>
> We first show any $k$ realizable by our input-level kernel (under Euclidean distance on some dataset $R$) can be constructed by applying the input-level kernel to our data $X$ under Mahalanobis distance with appropriately chosen $M$ matrix. Let $k$ be a realizable matrix - i.e. any desired positive semi-definite kernel matrix for which there exists a dataset $R'$ on which our input-level kernel satisfies $k = \phi(d(R', R')))$, where $d(\cdot, \cdot)$ denotes the matrix of Euclidean distances between all pairs of points in the first and second argument. Our construction first takes the entry-wise inverse $r=\phi^{-1}(k).$ Since $k$ is realizable, this $r$ must be a matrix of Euclidean distances between columns of a data matrix $R$ of the same dimensions as $X$. Assuming the gram matrix of $X$ is invertible, we simply solve the system $R=NX$ for a matrix $N,$  and set $M=N^TN,$ which yields $k=\phi(r)=\phi(d(R, R))=\phi(d_M(X, X)),$ where $d_M(\cdot, \cdot)$ denotes the operation that produces a Mahalanobis distance matrix for the Mahalanobis matrix $M$.
>
> We now give a construction demonstrating that the solution $k^*$ to problem $(2)$ is realizable up to arbitrarily small error using our input-level kernel on a dataset $R$ under Euclidean distance. We can realize the ideal kernel matrix that solves problem $(3)$, $yy^T$, up to arbitrarily small error by choosing $R$ according to a parameter $\epsilon > 0$, such that $\phi(d(R, R)) \rightarrow yy^T$ as $\epsilon \rightarrow 0$. In particular, for feature vectors $x_i,x_j$ of the same label in columns $i,j$ of $X$, we set the feature vectors $R_i, R_j$ for columns $i,j$ in $R$ to have $||R_i-R_j||=0$. Then, for $x_i,x_j$ in $X$ of different class, we set $R_i, R_j$ as columns of $R$ such that $||R_i-R_j|| > \epsilon^{-1}$. Then $k(R_i, R_j)$ is identically 1 for $R_i, R_j$ from the same class and converges to 0 for $R_i, R_j$ from different classes, as $\epsilon \rightarrow 0$, giving that $k = \phi(r) \rightarrow yy^T$.
>
> For any choice of $\epsilon>0$, we can apply the procedure described two paragraphs above to construct $M$ that realizes the same $k^*$ using our input-level kernel applied to our dataset $X$ under the Mahalanobis distance. Therefore, the solution to problem $(2)$ can be constructed as the infimum over feasible solutions to problem $(3)$, completing the proof.
>
> **Is there any easy way to square this with the results on RFMs, that the right singular values align with $\nabla f$ terms?**
>
> This is a keen observation. You are correct that, although the NFA describes structure in $W^T W$, the correlation between NFM and AGOP is enabled by the alignment between the left singular vectors of $W$ and the gradients with respect to the pre-activations, as argued in [1]. This can be seen through decomposing the AGOP as $W^T K W$, where $K$ is the covariance of $\frac{df}{dx}$ and $x$ are the pre-activations. Then, the NFA holds provided that the right singular structure of $W$ is not perturbed through the inner multiplication by $K$, an alignment that naturally occurs through training by gradient descent. See [1] for a more detailed description of this phenomenon.
>
> [1] Beaglehole, Mitliagkas, Agarwala. "Feature learning as alignment: a structural property of gradient descent in non-linear neural networks." arXiv pre-print, 2024.
>
> **It would be good to be explicit and put a subscript below the $\nabla$'s on the $\nabla f_\ell(x_{ci}^\ell)$**
>
> Thank you for this comment. We will fix this in the revision.
>
> **Why is Section 4.3 called “non-asymptotic analysis”?**
>
> We will rename this subsection “Connection to parametrized kernel ridge regression”.

---

### Author Rebuttal · Authors · 2024-08-01

We thank the reviewers for their thorough feedback on our manuscript. We will make a number of clarifying changes to the organization and presentation of our results. We list major changes here.

First, we will split Section 4 into two new sections. The first section will contain the empirical results in Subsection 4.1 together with the first row of Figure 3 in Appendix D of our original manuscript, which shows quantitative improvements in our DNC metrics for Deep RFM. We will then make Sections 4.2 and 4.3 into their own section, which will be renamed “Theoretical explanations of DNC in Deep RFM”.

Second, we will make the following improvements to the presentation of Section 4.2:
- We will first define $k_\Phi$ and $\lambda_\Phi$ at the end of the first paragraph.

- We will then add a paragraph following the first of Section 4.2 in our original manuscript to outline the argument and results of this subsection:
“We show that Deep RFM exhibits exponential convergence to DNC, and the convergence rate depends on the ratio $\lambda_{k} / \lambda_\Phi$. These  two parameters modulate the extent of DNC with Deep RFM. Namely, as we will show, if $k_{\mathrm{lin}}$ is close to the linear kernel, i.e., if $\lambda_{k}$ is small, then the predictor in each layer will resemble interpolating linear regression, an ideal setting for collapse through the AGOP. On the other hand, if $k_{\Phi}$ is close to the linear kernel, i.e., if $\lambda_\Phi$ is small, then the data will not be easily linearly separable. In that case, the predictor will be significantly non-linear in order to interpolate the labels, deviating from the ideal setting. We proceed by explaining specifically where $k_\Phi$ and $k_{\mathrm{lin}}$ appear in Deep RFM and why linear interpolation induces collapse, and then follow with the statement of our main theorem."

Third, we will do the following improvements to the presentation of Section 4.3:
- We will rename the subsection “Connection to parametrized kernel ridge regression" to clarify its role.

- We will rewrite the introduction of the subsection to clearly state its message: DNC is a natural consequence of minimizing the norm of the predictor over the choices of both the kernel function and the kernel coefficients in parametrized kernel ridge regression. We also clearly state why this result is relevant in our context – the RFM learning algorithm can be understood as an optimization procedure for the parametrized kernel ridge regression objective.

- We will discuss matrix-valued kernels in more detail and write the explicit formula for the kernel we are describing, as the reviewer uXDR suggests.

- We will discuss in more detail both why it is important to introduce $M$ and why we are able to consider the relaxed optimization over kernel matrices $k$. In particular, introducing $M$ is necessary as that is the key added parameter in the parametrized kernel ridge regression and it corresponds to the AGOP matrix used in RFM. Dropping the $M$ and optimizing over $k$ does not always lead to an equivalent optimization, as pointed out by reviewer AtBe, however their solutions are equivalent in the setting of our asymptotic analysis - where the data gram matrix is invertible. We explain this point in detail in our response to reviewer AtBe. Note that the invertibility assumption is guaranteed to hold when we consider collapse beyond the first layer of Deep RFM, at which we have inflated the data dimension through a high-dimensional feature map $\Phi_{\mathrm{lin}}$. For more detail on this point, see our response to reviewer Ljpy.

- We will call the problem $(3)$ “unconstrained kernel model.”

Fourth, we will do the following changes to Section 5:

- We will rename the section “AGOP as an effect of linear denoising in neural networks”, and emphasize the role of AGOP as responsible for improvements in NC1 in neural networks. In particular, we will rewrite the first paragraph as follows:
"We provide evidence that the DNC mechanism of Deep RFM, i.e., the projection onto the AGOP, is responsible for DNC formation in typical neural networks, such as MLPs, VGG, and ResNet trained by SGD with small initialization. We do so by demonstrating that DNC occurs by this iterated linear mechanism through the right singular structure of the weights. As the NFM, which is determined by the right singular structure of $W$, is highly correlated with the AGOP, our results imply the AGOP is responsible for NC1 progression."

---

### Comment · Area_Chair_hkw9 · 2024-08-08

Dear reviewers,

could you have a look at the authors response and comment on them if you have done so, yet.

thanks in advance

your area chair

---

### Decision · Program_Chairs · 2024-09-25

**Decision:**

Accept (poster)

**Comment:**

The submission introduces a interesting mechanism for Deep Neural Collapse (DNC) using the average gradient outer product (AGOP).  Given that the main results apply both to a non-standard kernel method and a non-standard training algorithm, it is unclear what the implications of the results are for more well-known models and algorithms.  Personally, how one can use DNC to improve the non-deep learning based feature learning algorithm can be more interesting than the story in the paper now. (e.g. Only from this manuscript, I can't know the performance of DeepRFM ). Although reviewers raised concern of studying non-standard mechanism, from my personal perspective, this paper can be an interesting add to the machine learning community. Thus I vote for acceptance.